# Optogenetic control of mRNA condensation reveals an intimate link between condensate material properties and functions

Min Lee[1,5], Hyungseok C. Moon[2,5], Hyeonjeong Jeong[2,3], Dong Wook Kim[2], Hye Yoon Park [2,3] ✉ & Yongdae Shin [1,4] ✉

Biomolecular condensates, often assembled through phase transition mechanisms, play key roles in organizing diverse cellular activities. The material properties of condensates, ranging from liquid droplets to solid-like glasses or gels, are key features impacting the way resident components associate with one another. However, it remains unclear whether and how different material properties would influence specific cellular functions of condensates. Here, we combine optogenetic control of phase separation with single-molecule mRNA imaging to study relations between phase behaviors and functional performance of condensates. Using light-activated condensation, we show that sequestering target mRNAs into condensates causes translation inhibition. Orthogonal mRNA imaging reveals highly transient nature of interactions between individual mRNAs and condensates. Tuning condensate composition and material property towards more solid-like states leads to stronger translational repression, concomitant with a decrease in molecular mobility. We further demonstrate that β-actin mRNA sequestration in neurons suppresses spine enlargement during chemically induced long-term potentiation. Our work highlights how the material properties of condensates can modulate functions, a mechanism that may play a role in fine-tuning the output of condensate-driven cellular activities.

Biochemical activities driving cell proliferation and survival are spatially organized through compartmentalization. Diverse types of membrane-less assemblies, also called biomolecular condensates, coexist within cells to concentrate a set of specific biomolecules and are thought to perform dedicated cellular functions[1,2]. In particular, various condensates are found along the flow of genetic information from transcription and RNA processing to translation[3-6]. Recent studies have shown that phase separation mechanisms drive the formation of condensates with compositions distinct from the surrounding protoplasm[7-9]. A network of transient multivalent intermolecular interactions, mediated by tandem interaction domains[10] or intrinsically disordered regions (IDR)[9,11], dictates the internal organization and composition of condensates[12]. Condensates are thought to function through multiple mechanisms, for example, by increasing the concentrations of specific enzymes and substrates to facilitate biochemical reactions[13], or by modulating the molecular availability and accessibility through sequestration of relevant factors[14,15]. However, the detailed mechanism of how specific functions can emerge from collective interactions of condensate components is still largely unclear.

As physical entities, condensates provide a local microenvironment for resident biomolecules. Physical properties of

[1]Interdisciplinary Program in Bioengineering, Seoul National University, Seoul, Korea. [2]Department of Physics and Astronomy, Seoul National University, Seoul, Korea. [3]Department of Electrical and Computer Engineering, University of Minnesota, Minneapolis, USA. [4]Department of Mechanical Engineering, Seoul National University, Seoul, Korea. [5]These authors contributed equally: Min Lee, Hyungseok C. Moon. ✉e-mail: hyp@umn.edu; ydshin@snu.ac.kr

condensates, such as density, material state, and viscosity, likely influence the way condensate components explore and interact with one another, which would ultimately impact condensate functions[16–18]. Typically, biomolecular condensates are viscoelastic network fluids composed of associative polymers[19]. Thus, depending on relevant timescales, condensates can behave as viscous liquids or elastic solids[20,21]. Different material properties may facilitate certain functionalities more effectively than others. For example, solid-like structures may perform better in the sequestration-based functional mode, but may not ideally work as reaction centers. These considerations suggest that the material properties of condensates may be selected for specific cellular functions and can be a target of regulatory mechanisms. Indeed, aberrant changes in the material properties of condensates are implicated in disease states including cancers[17].

The functional outcome of condensate formation as well as the effect of material properties on condensate activity have been primarily investigated using purified model systems. A recent work demonstrated that the co-condensation of SUMOylation enzymes and substrates accelerated the SUMOylation rate[22]. Synthetic DNA condensates also exhibited similar acceleration effects in strand displacement reactions[23]. Notably, the latter system highlighted an intimate link between the diffusive mobility of reactants and resulting reaction rates. Condensate-mediated compartmentalization can also suppress biochemical reactions; in-vitro protein droplets of FMRP and CAPRIN1[24], or elastin-like polypeptides[25] exhibited translation inhibitory activities. In contrast to these purified systems, probing how condensation contributes to specific cellular functions within living cells is a highly challenging task. Domain truncations or knockdown/out of phase-separating proteins may cause convoluted cellular effects in addition to simply perturbing condensation, thus identifying a causal relationship is often difficult[26]. In this regard, optogenetic tools enabling dynamic control of condensation can be a promising approach to dissecting the causal effect of intracellular condensation[27–29].

Here, we study the functional consequence of condensate formation by combining optogenetic techniques to control intracellular phase separation with single-molecule mRNA imaging. Light-induced sequestration of the reporter mRNAs into condensates leads to a decrease in the translational output from the resident mRNAs. Single-molecule imaging reveals that individual mRNA molecules often exhibit transient interactions with condensates. Rationally exploiting the phase behaviors of light-activatable components, we demonstrate that condensate composition and material property can be precisely modulated. Utilizing this capability, we show that solidifying condensates suppresses the mobility of resident molecules, at the same time further repressing translational activity from the sequestered mRNA. We also demonstrate that downstream cellular activities can be modulated by using condensate-based translation inhibition in neuronal cells. Our findings suggest that fine-tuning the material properties of condensates can be an important regulatory mechanism for proper cell physiology.

## Results

### Light-activatable biomolecular condensates sequester specific mRNAs

To study how biomolecular condensation can give rise to functions in living cells, we sought to reconstitute minimal synthetic condensates using a well-characterized light-inducible phase separation system, optoDroplet[27]. It has been shown that upon blue light exposure, the optoDroplet construct becomes activated to form large clusters through phase separation. To confer a cellular function to the optoDroplet condensate, we fused to the optoFUS (Cry2-mCh-FUS$_N$) the tandem MS2 coat protein (stdMCP) to recruit target mRNA species containing the bacteriophage MS2 binding sites (MBS)[30,31] (Fig. 1A). We chose to use an IDR from FUS protein (1-214; FUS$_N$) since unlike many

other IDRs, FUS IDR lacks the capacity to bind to RNA[32] while exhibiting strong self-association[11,27].

We first expressed optoMCP-FUS in mouse embryonic fibroblasts (MEFs) with 24 MBS loops inserted in the 3′ untranslated region (UTR) of their endogenous β-actin gene[31]. Upon blue light exposure, we observed that optoMCP-FUS rapidly formed clusters throughout the cell (Fig. 1B, C and Supplementary Movie S1). When the blue light was withdrawn, the optoMCP-FUS clusters dissolved back into the diffusive state (Supplementary Movie S2). To examine whether target mRNAs were recruited as designed, we performed single-molecule RNA fluorescence in situ hybridization (smRNA FISH) targeting the linkers between individual MBS stem-loops. Indeed, we found that the majority of MBS-tagged β-actin mRNA molecules (83% on average) were recruited into the optoMCP-FUS condensates (Fig. 1D, E). In contrast, the colocalization between the mRNAs and condensates was completely abolished in cells expressing optoFUS lacking the MBS binding capacity (Fig. 1D, E). The majority of optoMCP-FUS condensates contained single β-actin mRNA molecules, with the relatively low abundance of condensates exhibiting up to ~10 target mRNAs (Supplementary Fig. 1A and B). In contrast to β-actin, endogenous GAPDH mRNAs, devoid of the MBS, remained outside of the optoMCP-FUS condensates (Supplementary Fig. 1C). Thus, our optoMCP-FUS system can sequester target mRNAs through specific interactions between the MBS stem-loop and MCP.

We then examined the clustering behavior of optoMCP-FUS in detail. Given the strong binding affinity between MCP and MBS (K$_d$ < 1 nM)[33], we reasoned that individual β-actin mRNA molecules could be identified as distinct puncta at relatively low levels of MCP expression, a condition typically used for single mRNA imaging[34,35]. Indeed, in MEF cells expressing the low concentration of optoMCP-FUS, we were able to observe individual complexes of optoMCP-FUS and β-actin mRNA (Supplementary Fig. 2A and Supplementary Movie S3). As expected, the diffusive mobility of individual β-actin mRNA complexes was much faster than the optoMCP-FUS condensates (Supplementary Fig. 2B). When activated with blue light, clustering behaviors were not observed in these cells with the low levels of optoMCP-FUS (Supplementary Fig. 2A), suggesting that the cellular optoMCP-FUS concentrations were below a saturation concentration for phase separation[27]. To quantify the saturation concentration of optoMCP-FUS, we examined many cells expressing a broad range of optoMCP-FUS for their capacity to form light-activated condensates. We found that there exists a clear concentration threshold segregating cell populations for clustering (Fig. 1F). Notably, individual mRNA-optoMCP-FUS complexes became hardly identifiable in cells near or above the saturation concentration due to high background from unbound optoMCP-FUS species (Supplementary Fig. 2A). We then sought to probe whether the presence of tethered mRNAs affects the saturation concentration of optoMCP-FUS. This is motivated by earlier studies reporting that IDR oligomerization lowers saturation concentration, and potentiates phase separation[27,36]. Consistent with this view, we observed that in the presence of the MBS-tagged mRNA which can act as oligomerization scaffolds, the saturation concentration of optoMCP-FUS became lowered by ~20% (Fig. 1F). The decrease in saturation concentration also manifested as stronger clustering behaviors in cells with cognate mRNAs (Fig. 1G, H).

During light activation, optoMCP-FUS clusters often grew in size through fusing with one another (Fig. 1I), as typically observed for phase-separated condensates[37,38]. To examine molecular mobility within optoMCP-FUS condensates, we performed fluorescence recovery after photobleaching (FRAP) experiments. We found that optoMCP-FUS condensates exhibited a significant degree of fluorescence recovery (Fig. 1J). Together with the shape relaxation observed during fusion, these data suggest that optoMCP-FUS condensates behave like liquid droplets. Taken together, we demonstrate that using

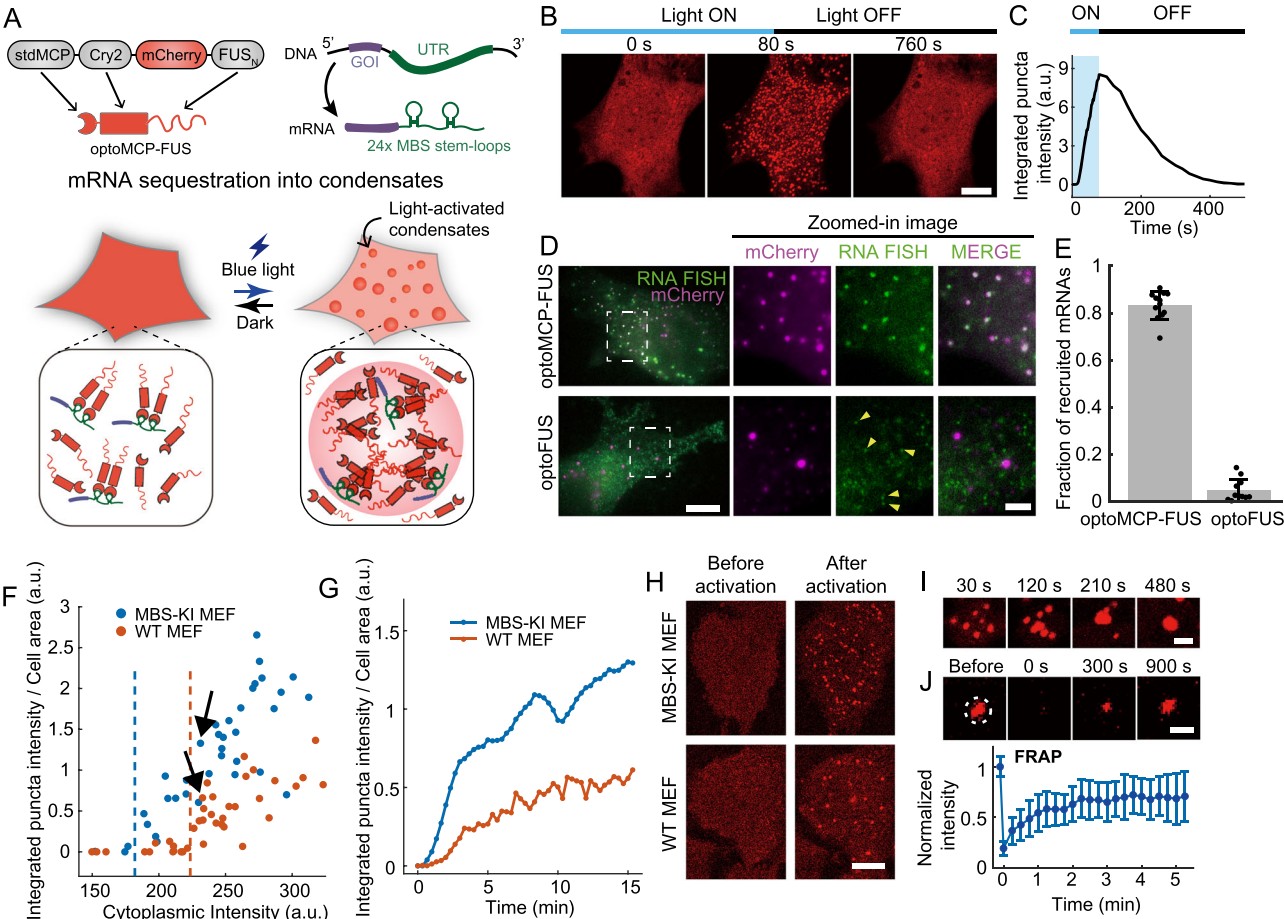

**Fig. 1 | Light-activated condensates sequester specific mRNAs. A** Schematic diagrams of the optoMCP-FUS system. optoMCP-FUS consists of stdMCP, Cry2PHR, mCherry and the N-terminal IDR of FUS. Upon blue light exposure, optoMCP-FUS undergoes phase separation to form condensates that specifically recruit the MBS-tagged mRNAs. **B** Representative confocal images of the MEF cells expressing optoMCP-FUS during blue light activation and deactivation. The cell is imaged and activated every 2 s. See also Supplementary Movie S1 and S2. **C** Temporal evolution of the integrated fluorescence intensity of light-activated condensates shown in (**B**). **D** Two-color fluorescence images of fixed MEF cells, with the MBS-tagged β-actin gene, expressing optoMCP-FUS (top) and optoFUS (bottom). The MBS-tagged β-actin mRNAs, visualized with smFISH (green), and light-activated condensates (magenta) are shown. Arrowheads indicate individual β-actin mRNA molecules. **E** Fraction of mRNAs recruited into the light-activated condensates. After 20 min of blue light activation, the recruitment of mRNAs into light-activated condensates was quantified from smFISH data. $n = 12$ cells for

optoMCP-FUS and $n = 11$ for optoFUS cells. Data are mean ± SD. **F** The degree of condensate formation as a function of the expression level of optoMCP-FUS. For individual MEF cells, the integrated intensity of the optoMCP-FUS condensates after 15 min of blue light activation is plotted against the initial cytoplasmic intensity of optoMCP-FUS. Arrows indicate individual cells shown in (**G**) and (**H**). **G** Temporal evolution of the integrated intensity of optoMCP-FUS condensates after blue light exposure. **H** Confocal images of MBS-KI (top) and WT MEF cell (bottom) expressing optoMCP-FUS. **I** Time-lapse confocal images showing fusion events between optoMCP-FUS condensates. **J** Confocal images of an optoMCP-FUS condensate (top) and the fluorescence recovery curve (bottom) in FRAP experiments. Cells were activated every 15 s to induce phase separation. A white dashed circle indicates a bleached area. $n = 14$ cells. Data are mean ± SD. Scale bars, 10 μm (**B**, **D**, and **H**), 2 μm (zoomed-in image in **D** and **I**), and 1 μm (**J**). a.u., arbitrary units. Source data for panels **C**, **E**–**G**, and **J** are provided in the Source Data file.

the light-activatable optoMCP-FUS system, specific target mRNAs can be recruited into phase-separated intracellular condensates.

## Live-cell mRNA imaging shows dynamic interactions between optoMCP-FUS condensates and mRNAs

We then sought to probe how individual mRNA molecules interact with optoMCP-FUS condensates. We noticed that at high levels of optoMCP-FUS expression above the saturation concentration, the high background of unbound optoMCP-FUS in the cytoplasm prevents direct visualization of individual mRNA molecules (Supplementary Fig. 2A). To overcome this limitation, we employed a second, ortho-gonal RNA-tagging system consisting of the PP7 binding site (PBS) and the PP7 coat protein (PCP)[39]. Previous studies used the PBS-PCP system simultaneously with the MBS-MCP to label different RNA species[40]. We thus tagged a gene of interest (GOI) with 12 repeats of MBS and PBS stem loops at its 3′UTR (Fig. 2A). When used together with optoMCP-

FUS and tandem PCP fused with tandem GFP (stdPCP-stdGFP), the tagged mRNAs can be localized in the GFP channel, independent of the light-induced condensation imaged in the mCherry channel. The nuclear localization signal (NLS) was fused to the N-terminus of stdPCP-stdGFP to lower the cytoplasmic background fluorescence and enable single-molecule imaging of mRNAs (Supplementary Fig. 3A)[41,42].

We expressed optoMCP-FUS and stdPCP-stdGFP in MEF cells with 12 repeats of MBS and PBS inserted into the 3′UTR of endogenous c-Fos gene (Fig. 2A). To capture the physical interactions between individual mRNAs and optoMCP-FUS condensates, we paid special attention to choosing cells expressing suitable levels of both optoMCP-FUS and stdPCP-stdGFP (Supplementary Fig. 2A); if the expression level of optoMCP-FUS is too low, then no phase separation would be triggered; Additionally, too high levels of stdPCP-stdGFP would only increase the background fluorescence, preventing the visualization of mRNA molecules. From cells with appropriate levels of

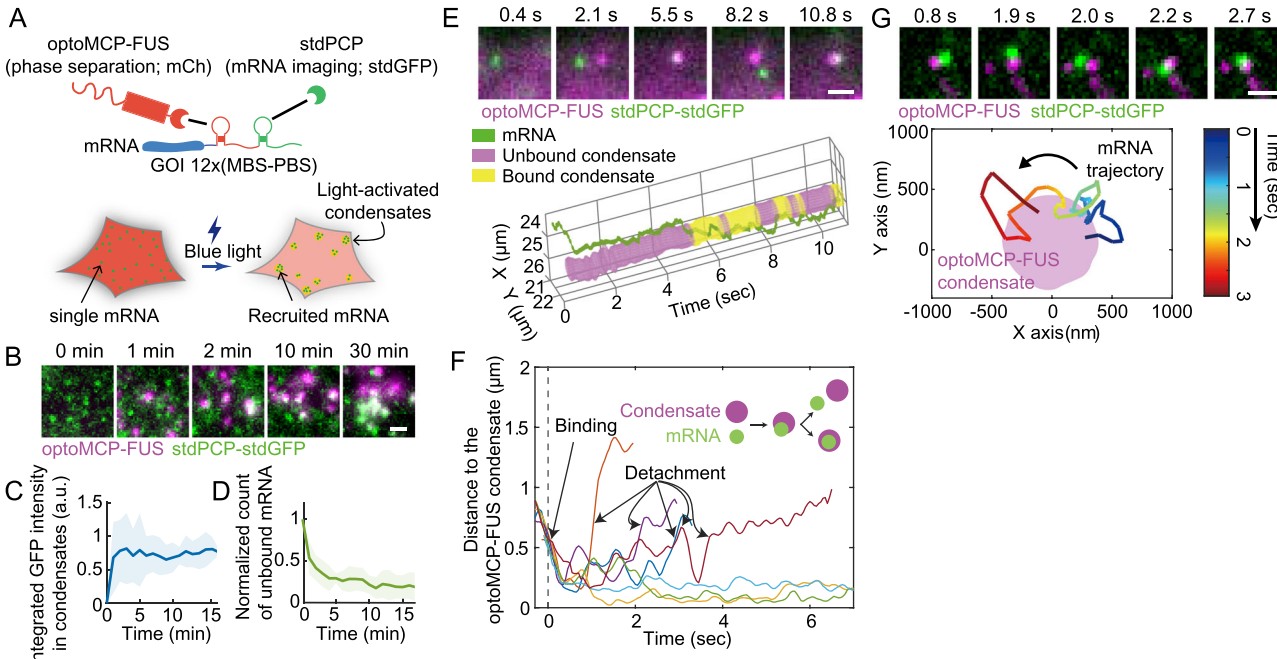

**Fig. 2 | Live-cell single mRNA imaging shows dynamic interactions between optoMCP-FUS condensates and target mRNAs. A** Schematic diagram of the mRNA dual labeling strategy for single mRNA imaging orthogonal to light-activated phase separation. 12 repeats of MBS-PBS loops were inserted at the 3'UTR of the gene of interest. Individual mRNAs can be imaged as distinct foci through the PBS-bound stdPCP-stdGFP even in the presence of the high background level of optoMCP-FUS. **B** Time-lapse fluorescence images of individual mRNAs, visualized with stdPCP-stdGFP, and optoMCP-FUS condensates in the MEF cell with the endogenous c-Fos gene tagged with MBS-PBS. The cell was imaged and activated every 1 min. Images are maximum-intensity projections, and post-processed through rolling-ball background subtraction, bleach-correction, and bead-based drift correction. **C** Time evolution of the integrated intensity of stdPCP-stdGFP within optoMCP-FUS condensates during blue light induction. Data are mean (solid line) ± SD (shaded area). **D** Normalized counts of the unbound mRNAs during blue light induction. Data are mean (solid line) ± SD (shaded area). *n* = 4 cells (**C, D**). **E** (Top) Example images of the optoMCP-FUS condensate interacting with the target mRNA. The MEF cell was imaged and activated every 50 ms after 30 s of blue light activation. Images were walking-averaged, bleach-corrected, and drift-corrected. See also Supplementary Movie S4. (Bottom) The trajectory of the mRNA and the optoMCP-FUS condensate shown in the images above. **F** Distances between individual mRNAs and the nearest optoMCP-FUS condensates were tracked for several target mRNAs. **G** (Top) Example images of the target mRNA molecule scanning the surface of the optoMCP-FUS condensate. The MEF cell was imaged and activated every 50 ms after 30 s of blue light activation. Images were walking-averaged, bleach-corrected, and drift-corrected. (Bottom) The mRNA trajectory is color-coded temporally during the scanning. Scale bars, 1 μm (**B, E,** and **G**). a.u., arbitrary units. Source data for panels **C–G** are provided in the Source Data file.

optoMCP-FUS and stdPCP-stdGFP, we were able to capture the dynamics of the target mRNA recruitment into the condensates (Fig. 2B). Prior to blue light exposure, individual mRNAs were visible in the GFP channel, but no puncta were observed in the mCherry channel due to the high background of free optoMCP-FUS. Blue-light activation triggered the formation of optoMCP-FUS condensates into which the reporter mRNAs became recruited. We found that the mRNA recruitment took place rapidly, reaching a saturation level in less than ~5 min (Fig. 2C, D).

To monitor fast molecular dynamics during the sequestration of reporter mRNAs, we increased the acquisition rate of two-color fluorescence data to 20 Hz (frames/sec). Consistent with results from optoMCP-FUS-based imaging (Supplementary Fig. 2A and B), the motility tracking of mRNAs and condensates showed that individual mRNA molecules diffused much faster than condensates (Supplementary Fig. 3B and C), even though their apparent sizes often appear to be similar due to the diffraction-limited resolution. Strikingly, we found that interactions between individual reporter mRNAs and the optoMCP-FUS condensates were often transient; once encountering each other, multiple events of short (<1 sec) binding and unbinding were frequently observed (Fig. 2E, F, and S3D and Supplementary Movie S4). In some cases, the reporter mRNA appeared to scan the surface of the optoMCP-FUS condensates while being loosely tethered (Fig. 2G). Interestingly, we also observed examples of stable associations between mRNAs and condensates (Fig. 2F and S3E and Supplementary Movie S5). These heterogenous behaviors of varying binding

stabilities are consistent with previous observations and may be attributable to the different translational status of individual mRNAs at the time of contact[43]. Thus, although individual interactions between mRNAs and condensates are often transient and reversible in short timescales, the accumulation of the optoMCP-FUS molecules onto the condensates appears to eventually stabilize these interactions and promotes the sequestration of the reporter mRNA over longer timescales.

## Sequestration of mRNA into the optoMCP-FUS condensates inhibits translation

We next asked how mRNA sequestration into the condensates would influence translational activity. To monitor the translation activity in live cells, we used tagBFP as the GOI in our MBS-PBS reporter system (Fig. 3A). We appended a destabilization domain (DD) tag[44] at the C-terminus of tagBFP to speed up its decay and thereby improve the response rate of tagBFP signals as an indicator for translation activity. In the absence of light-activated condensation, cells expressing higher concentrations of optoMCP-FUS tended to exhibit higher tagBFP levels (Supplementary Fig. 4A and B), indicating that binding of optoMCP-FUS to the 3'UTR may affect the reporter mRNA stability[45]. To examine the effects of mRNA sequestration into condensates on translation, we measured the temporal changes of tagBFP levels in individual U2OS reporter cells during 12 h of blue light activation. Although a high cell-to-cell variation in tagBFP fluorescence was observed, we found that the tagBFP intensities of individual cells progressively decreased

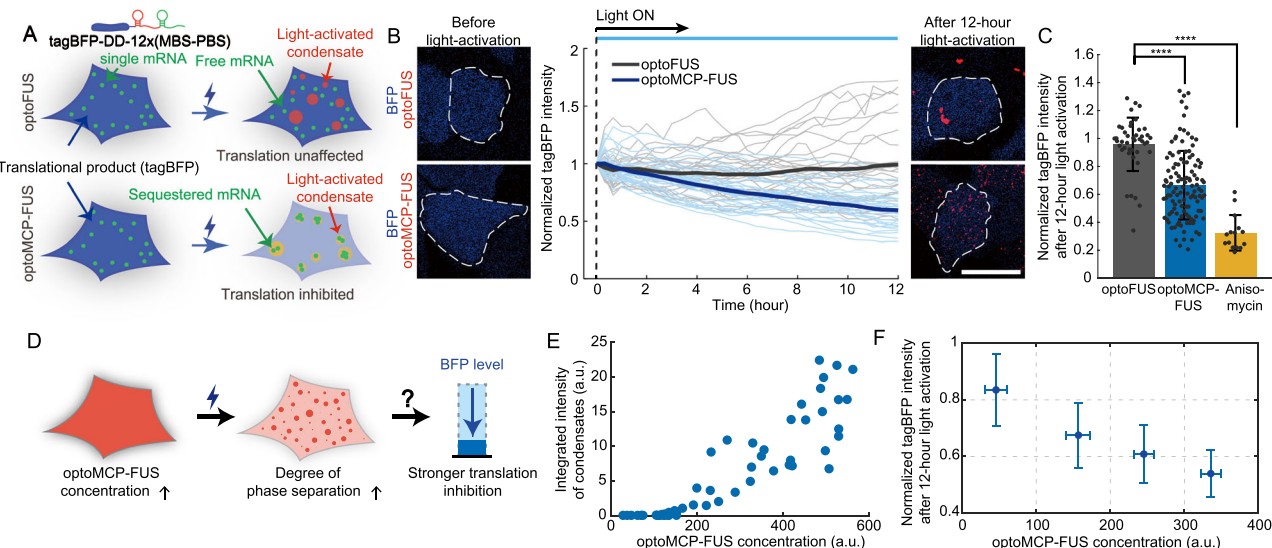

**Fig. 3 | Sequestering the target mRNAs into optoMCP-FUS condensates inhibits protein translation. A** Schematic diagram showing translation inhibition, monitored with tagBFP fluorescence, due to the mRNA sequestration into light-activated condensates. **B** During 12 h of blue light activation, tagBFP fluorescence levels were measured for individual U2OS cells with the tagBFP mRNA reporter (middle). The bold curves are averaged values. (right and left) Confocal images before and after light activation are shown. *n* = 32 (optoMCP-FUS) and 30 (opto-FUS). Cells were activated with filtered DIA light. **C** Normalized tagBFP intensity of individual cells after 12 h of light activation. The fluorescence intensities are normalized with the initial values of individual cells prior to light activation. *n* = 43

(optoFUS), 117 (optoMCP-FUS), and 15 (anisomycin). Statistical significance was calculated using Student's two-tailed *t* test. ****$P$ < 0.0001. $P$ = 4.71E-11 (optoFUS and optoMCP-FUS) and 5.38E-17 (optoFUS and anisomycin). Data are mean ± SD. **D** Schematic of the effect of the optoMCP-FUS expression level on the extent of translation inhibition. **E** Scatter plot for the integrated intensity of optoMCP-FUS condensates for individual cells with varying expression levels. **F** Normalized tagBFP intensity after 12 h of light activation as a function of optoMCP-FUS expression levels. *n* = 32 (0–100), 25 (100–200), 34 (200–300) and 12 cells (300–400). Data are mean ± SD. Scale bar, 10 μm (**B**). a.u., arbitrary units. Source data for panels **B, C, E,** and **F** are provided in the Source Data file.

during blue light activation (39% reduction in 12 h) (Fig. 3B). In contrast, control experiments with the optoFUS construct showed no decay in the amount of tagBFP under the identical activation and imaging conditions. Thus, the recruitment of mRNAs into the condensates has a functional consequence of inhibiting protein translation. We note that optoMCP-FUS condensation did not lead to either stress granule formation (Supplementary Fig. 5A) or global changes in protein production (Supplementary Fig. 5B).

The degree of translation inhibition observed in the optoMCP-FUS sample was not as high as one from the sample treated with anisomycin, a protein synthesis inhibitor (Fig. 3C). Since in our optoMCP-FUS system, the inhibition was achieved through light-induced recruitment of mRNAs into condensates, we reasoned that translation inhibition might depend sensitively on the concentration of optoMCP-FUS (Fig. 3D). Indeed, we found that the total volume of light-activated condensates in each cell strongly depended on the cytoplasmic concentration of optoMCP-FUS (Fig. 3E), and that higher optoMCP-FUS concentrations caused stronger translation inhibition (Fig. 3F). This behavior may also partially explain the origin of the high cell-to-cell variation in translation inhibition observed at the population level (Fig. 3B).

## Modulating the material properties of condensates influences their functions

Another factor that can impact the inhibition process is the material properties of condensates. Depending on the material state, biomolecules residing in condensates experience different local environments[46]; in the liquid-like state, they are involved in dynamic interactions with neighboring molecules which constantly undergo rearrangements; in contrast, biomolecules in the solid-like state tend to be caged by nearby molecules exhibiting arrested local dynamics. Using our optoMCP-FUS condensates as a reconstituted minimal model, we sought to directly probe how the change in the material state would modulate condensate functionalities (Fig. 4A). To control

material property, we took a strategy of tuning condensate composition, following the concept of ternary regular solution systems[47] (Supplementary Fig. 6A). Specifically, optoFUS construct (FUS_N-miRFP670-Cry2), previously shown to form solid-like gels[27] (Supplementary Fig. 6B and C), was co-expressed in optoMCP-FUS cells. Since in these two constructs, identical motifs mediate intermolecular interactions driving phase separation (FUS_N-FUS_N as well as Cry2-Cry2), we reasoned that co-condensation would give rise to condensates enriched in both constructs (Fig. 4A and Supplementary Fig. 6A), as well as with reduced internal dynamics.

When experimentally tested, we indeed found that the phase diagram of optoMCP-FUS and optoFUS qualitatively matched with what was expected from the ternary regular solution model (Fig. 4B and C, and Supplementary Fig. 6A); Instead of forming two immiscible dense phases, a single dense phase enriched in both constructs constituted light-activated condensates. We then probed how condensate composition was modulated. To quantify condensate composition, we measured fluorescence intensities of light-activated condensates for a broad range of expression levels of both constructs. Since (1) optoMCP-FUS condensates tended to be small and (2) puncta intensities were prone to underestimation for small puncta, we used a ratio of optoFUS fluorescence to optoMCP-FUS as a proxy for condensate composition (Fig. 4D). We found that depending on the relative locations in the phase diagram, cells of optoMCP-FUS and optoFUS exhibited distinct molecular compositions (Fig. 4D and E, and Supplementary Fig. 6D and E), which is consistent with the predictions from the ternary regular solution model[48] (Supplementary Fig. 6A). Cells with higher concentrations of optoFUS exhibited light-induced condensates enriched relatively higher in optoFUS, and vice versa (Fig. 4D, E, and Supplementary Fig. 6D, E).

Having established a way to systematically modulate condensate composition, we then investigated the effect of different compositions on the degree of translation inhibition. Using the blue-light activation protocol identical to the one applied to optoMCP-FUS alone (Fig. 3B),

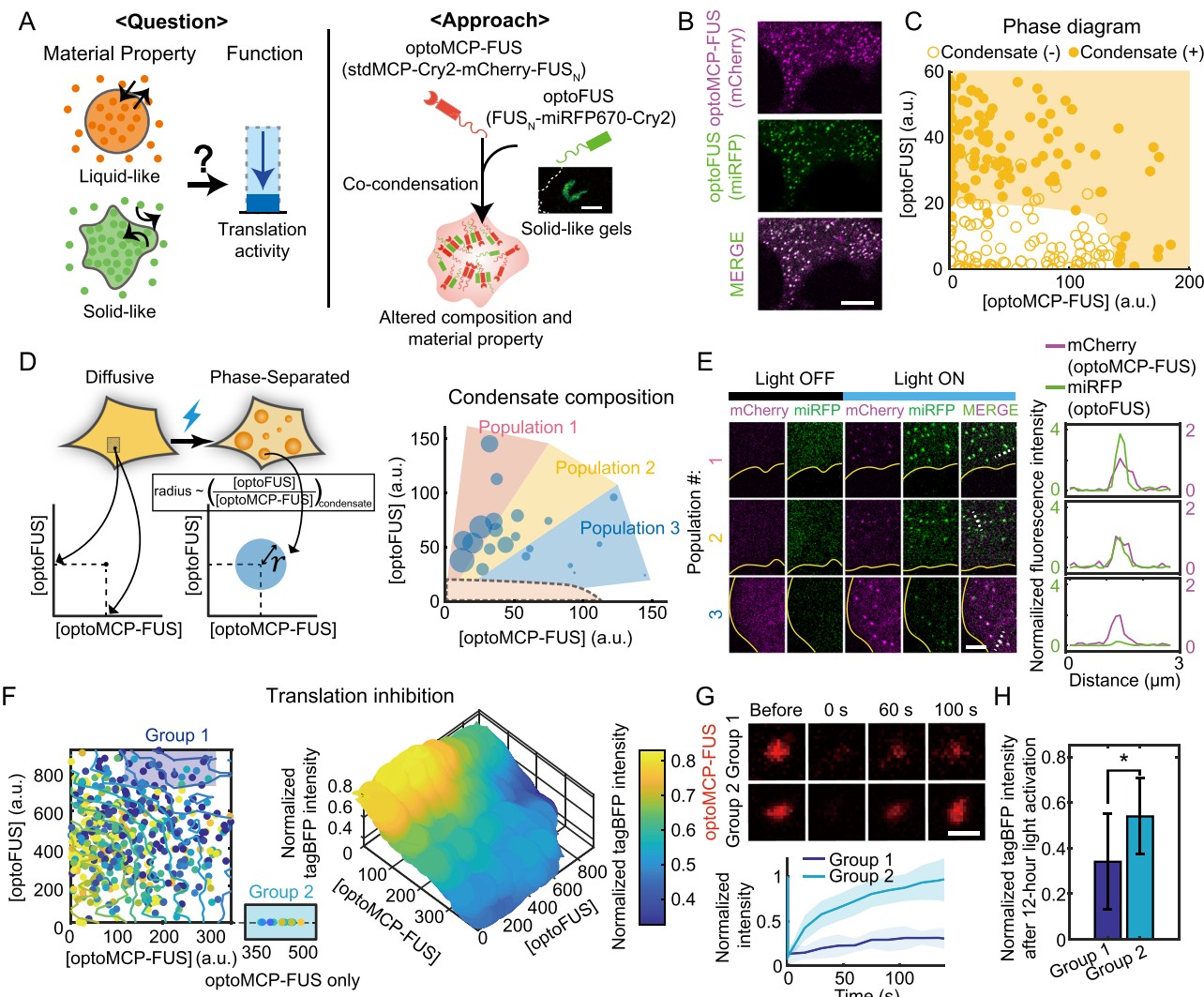

**Fig. 4 | Solidifying mRNA-containing condensates strengthens translation repression. A** (Left) Schematic of the effect of different material properties on condensate function. (Right) Condensate composition is altered to accommodate optoFUS, a solid-like gel-forming light-activatable component. Inset: an image of the gel-like optoFUS cluster in a U2OS cell. **B** Example confocal images of the light-activated U2OS cell expressing the tagBFP reporter mRNA, optoMCP-FUS, and optoFUS. **C** Phase diagram of blue-light activated U2OS cells expressing optoMCP-FUS and optoFUS. Solid circles indicate cells with condensates while empty circles are those without condensates. **D** (Left) For individual cells, measured condensate compositions were visualized as circles of different radii. (Right) The measured composition of condensates in cells with different levels of optoMCP-FUS and optoFUS. A region with a dashed boundary near the origin corresponds to the concentration range where no phase separation was observed as in (**C**). **E** (Left) Representative confocal images of cells from each population in (**D**). Cell boundaries are indicated with yellow curves. (Right) Intensity profiles of optoMCP-FUS

and optoFUS along white dashed lines. Fluorescence intensities were normalized with the maximum pixel values of optoMCP-FUS in each condensate. **F** (Left) Scatter plot for the normalized tagBFP intensities of individual cells after 12 h of blue light activation. A group of cells with strong translation repression, labeled as group 1, was chosen for further analysis. Cells in group 2, expressing only optoFUS, were chosen to have similar levels of clustering as those in group 1 (Supplementary Fig. 6F). (Right) 3D contour graph of the moving-averaged scatter plot. **G** (Top) Fluorescence images of condensates in two different groups in (**F**) during FRAP experiments. (Bottom) Fluorescence recovery curves for condensates in each group. Data are mean (solid line) ± SD (shaded area). $n = 12$ (Group 1) and 11 (Group 2). **H** Normalized tagBFP intensity after 12 h of light activation for cells in group 1 and 2. Data are mean ± SD. $n = 18$ (Group 1) and 12 (Group 2). *$P < 0.05$. $P = 0.0102$. Scale bars, 10 μm (**A** and **B**), 5 μm (**E**), and 1 μm (**G**). a.u., arbitrary units. Source data for panels (**C**, **D**, and **F–H**) are provided in the Source Data file.

translation inhibition was measured for individual cells based on the remaining tagBFP fluorescence. We observed that in general, cells exhibited a strong cell-to-cell variation in the extent of translation inhibition (Fig. 4F), yet two interesting features were noticeable. First, consistent with the results from the optoMCP-FUS alone (Fig. 3F), the higher concentration of optoMCP-FUS tended to suppress translation more strongly. More importantly, in a region of the ternary phase diagram where the condensate composition is expected to be biased toward the high fraction of optoFUS (labeled as group 1 in Fig. 4F), we observed stronger translation inhibition compared to cells in other regions. Considering the high doping ratio of optoFUS within the

condensates, we speculated that the condensates in this group were likely to be overall more solid-like, compared to those in optoMCP-FUS only cells. To test this idea, we quantified the molecular mobility and translation inhibition for cells in group 1 and then compared them to the optoMCP-FUS alone cells (group 2). In doing so, cells in group 2 were chosen so that the total amounts of condensates were similar between the two groups (Supplementary Fig. 6F). Remarkably, we found that condensates in group 1, with a high fraction of optoFUS, were indeed more solid-like as indicated by a higher immobile fraction in FRAP experiments (Fig. 4G), and exhibited stronger translation inhibition (Fig. 4H). Group 2 showed stronger recruitment of the

reporter mRNAs (Supplementary Fig. 6G and H), further supporting the solid-like material property as a key mechanism for the observed translation suppression. Taken together, using our light-activatable condensates, we find direct experimental evidence linking the condensate material properties to functions. In particular, our results show that solidifying condensates can repress translation from residing mRNAs.

### Translation inhibition by mRNA sequestration occurs in short timescales

Although we used the destabilization domain (DD) tag to accelerate protein degradation, the response speed of the translation reporter system is fundamentally limited by the decay rate of tagBFP which was ~6 h (half-life) in our study (Supplementary Fig. 7). To investigate whether translation inhibition is effective immediately after light-activated condensation, we employed a puromycylation-PLA (Puro-PLA) assay which utilizes proximity ligation to localize specific polypeptides newly synthesized during a short incubation time in puromycin[49]. In the assay, a low concentration of puromycin is introduced to label and release elongating polypeptide chains from ribosomes which are then recognized by two different antibodies targeting specific protein of interest and puromycin, respectively.

We first expressed optoMCP-FUS in MEF cells with MBS-labeled β-actin mRNA, and activated them with blue light for 20 min to induce condensate formation and mRNA sequestration (Supplementary Fig. 8A). While maintaining blue light activation, cells were then treated with puromycin for 5 min during which nascent polypeptides were primed for amplification and detection in the Puro-PLA assay. When compared to a control sample without any blue light activation, we found that condensation led to a decrease in the number of Puro-PLA puncta by 28% (Supplementary Fig. 8B and C). Thus, our results indicate that translation inhibition from condensate-based mRNA sequestration becomes effective as early as 20 min after light activation. Notably, no changes in the total amount of target mRNAs were observed during light activation (Supplementary Fig. 8D and E). This result further confirms that the decrease in translational output is due to spatial redistribution of mRNAs, i.e., mRNA sequestration into condensates, rather than other processes affecting their abundance.

### Perturbation of endogenous β-actin mRNA translation during cLTP in live neurons

We then sought to investigate whether condensate-based translation inhibition can modulate downstream cellular functions. For this purpose, we chose to probe the process of spine enlargement during the chemical long-term potentiation (cLTP) stimulation. Dendritic spines are small protrusions from the dendrites of neurons where the postsynaptic sites of excitatory synapses are located. Upon synaptic stimulation, some dendritic spines exhibit a long-lasting volume increase for more than an hour[50], which is referred to as structural long-term potentiation (sLTP)[51,52]. This structural plasticity is highly dependent on the dynamics of actin cytoskeleton[53,54], and associated with mRNA localization and local translation of proteins[52,55,56]. In this regard, we applied our optoMCP-FUS system to transiently block β-actin translation with blue light, and investigated the effect of this altered translation on the structural responses of individual dendritic spines during cLTP stimulation.

We first expressed optoMCP-FUS in dissociated hippocampal neurons with endogenous β-actin gene labeled with MBS at the 3'UTR (Actb-MBS neurons)[31], and performed a series of characterization of light-activated condensation. When activated with blue light, these neurons exhibited global clustering of optoMCP-FUS, as observed in MEF cells (Fig. 5A). Since neurons have highly extended structures, we wondered if localized condensation was readily achievable using the optoMCP-FUS system. When a highly localized light activation was applied to a subcellular region of a neuron, we found that a single optoMCP-FUS cluster formed at the illuminated region (Fig. 5B). We then sought to probe whether target mRNAs are effectively sequestered into light-activated optoMCP-FUS condensates in neurons. The smRNA FISH experiments showed that endogenous β-actin mRNAs were recruited into the optoMCP-FUS condensates in the neuronal dendrites (Fig. 5C and Supplementary Fig. 9). Thus, our optoMCP-FUS system allows for controlled formation of condensates with targeted sequestration of specific mRNA species in neurons.

We then asked whether light-induced sequestration of β-actin mRNA into condensates would influence the sLTP (Fig. 5D). When cLTP stimulation was applied to MBS-KI neurons in the absence of blue light, we found that the spine volume increased over time (Fig. 5E, F), as previously reported[56,57]. However, in the blue-light activation condition, a volume increase in the dendritic spines was significantly suppressed (Fig. 5E). Compared to the dark condition, we observed a 42% reduction in the average volume increase in the light-induced mRNA condensation (Fig. 5F and G). Consistent with the requirement of localized translation for the spine growth, cycloheximide (CHX) treatment caused strong suppression of sLTP (Fig. 5F and G). To further test the molecular specificity of the condensate-induced sLTP inhibition, we applied identical cLTP stimulation to wildtype neurons, without the integrated MBS in the β-actin gene, in the presence of blue light condition. This control sample showed the spine growth similar to the dark condition (Fig. 5F and H), indicating that blue light itself was not the origin of the observed sLTP inhibition. Taken together, these results demonstrate that our light-inducible condensate system can sequester specific endogenous mRNA species in neurons, which can modulate localized cellular activities such as the spine enlargement associated with synaptic plasticity.

## Discussion

Our work directly shows that mRNA sequestration into biomolecular condensates has a functional consequence of decreasing translational output from resident mRNAs. Without invoking changes in the sequence or expression level of regulatory proteins, we use the optogenetic approach to control the intracellular phase of engineered RNA-binding proteins which specifically associate with target mRNAs. We find that condensation of the target mRNA leads to translation repression as early as 20 min after light activation, as shown in Puro-PLA experiments. Consistent with condensation-driven behaviors, we observe a stronger repression in cells expressing higher concentrations of the phase-separating component. Interestingly, we find that the degree of translation repression is lesser than what is expected based on the level of mRNA sequestration. We attribute this behavior to the highly transient nature of interactions between individual mRNAs and condensates, as observed in our single-molecule tracking experiments. Thus, individual mRNAs targeted to condensates likely experience a stochastic competition between translation initiation/elongation and sequestration into condensates. Further employment of single-molecule imaging of translation activities[58-61] can help elucidate the fate of mRNAs localized to biomolecular condensates in different cellular contexts.

RNA is central to the organization of diverse condensates by engaging in various RNA-protein and RNA-RNA interactions[18,62]. For functional aspects, partitioning of mRNAs into condensates is associated with heterogeneous outcomes, ranging from translational induction to inhibition[63]. In contrast to a classical view of RNP condensates concentrating translationally-repressed mRNAs[64], recent studies involving simultaneous imaging of mRNA localization and translation activity have identified several condensates harboring translationally active mRNAs. These so-called translation factories are enriched with specific mRNAs including those encoding glycolytic enzymes[65] and signaling effectors[66]. In addition, during spermiogenesis in mice, liquid-liquid phase separation of FXR1 and target mRNAs important for spermatid development was shown to be indispensable

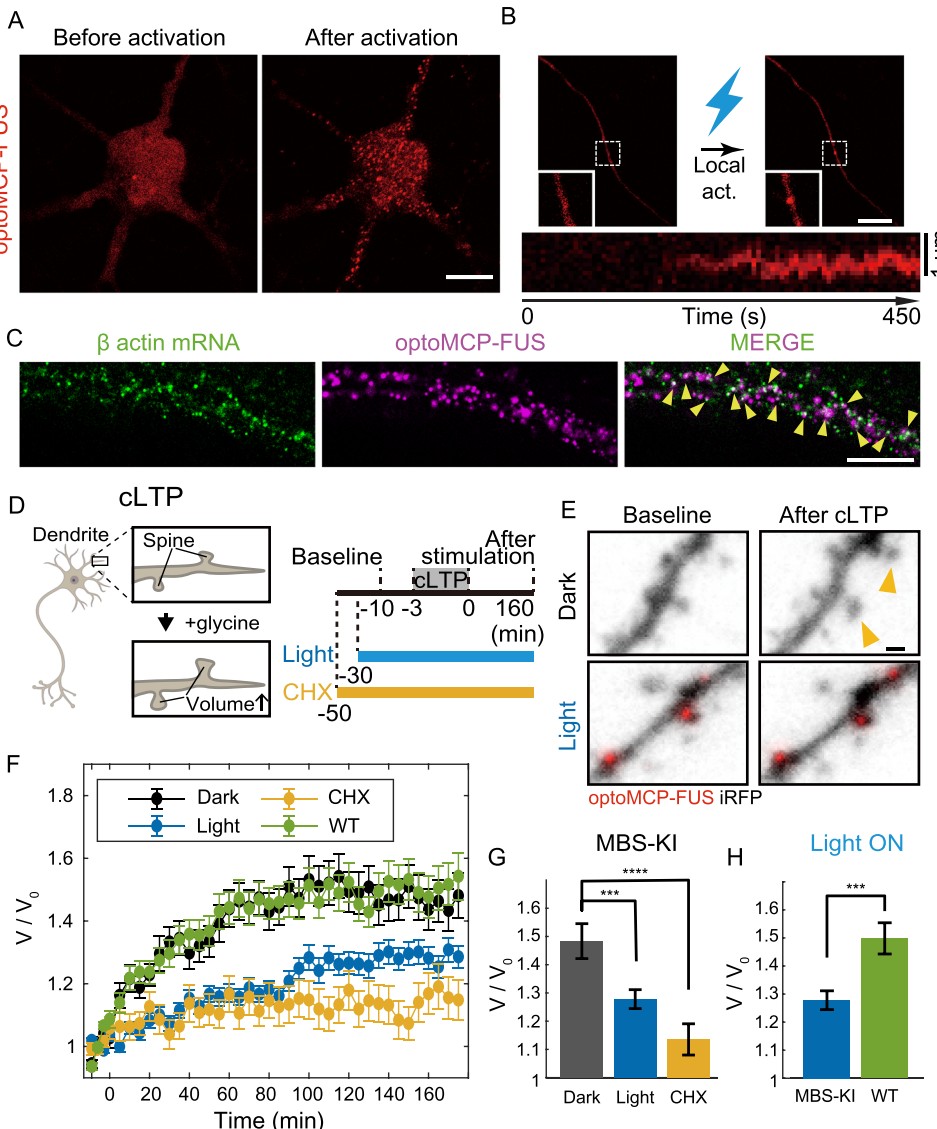

**Fig. 5 | Perturbation of endogenous β-actin mRNA translation during chemical LTP in live neurons. A** Confocal images of the MBS-KI neuron showing optoMCP-FUS condensate formation. Cells were activated every 2 s. **B** Confocal images of localized condensate formation in the MBS-KI neuron. A subcellular region of the neuron was locally activated with blue light (Top). (Bottom) A kymograph was generated along a horizontal line crossing the center of the optoMCP-FUS condensate (drift-corrected). Cells were activated every 2 s. **C** Two-color fluorescence images of a fixed optoMCP-FUS expressing neuron with the MBS tagged β-actin gene. Yellow arrowheads indicate co-localization between β-actin mRNAs and optoMCP-FUS condensates. **D** (Left) The spine enlargement during the chemical long-term potentiation (cLTP). (Right) Time lines of cLTP perturbation experiments. **E** Representative images of dendrites during cLTP with and without blue light activation. Yellow arrowheads indicate enlarged dendritic spines. **F** The average spine volume change over time after cLTP. $n = 174$ (Dark), 307 (Light), 167 (WT), and 128 spines (CHX). Data are mean ± s.e.m. **G** Quantification of average spine size 160 min after cLTP stimulation in MBS-KI neurons. $n = 174$ (Dark), 307 (Light), and 128 spines (CHX). $p = 3.52E\text{-}05$ (Dark and CHX) and 7.61E-04 (Dark and Light). Data are mean ± s.e.m. **H** Quantification of average spine size 160 min after cLTP stimulation in the light activation condition. $n = 307$ (Light) and 167 spines (WT). $p = 1.77E\text{-}04$. Data are mean ± s.e.m. ****$P < 0.0001$ and ***$P < 0.001$. Scale bars, 10 μm (**A**–**C**) and 1 μm (**E**). Source data for panels (**F**–**H**) are provided in the Source Data file.

for activating protein translation[67]. Translation initiation factors such as EIF4G3 are recruited into FXR1 condensates to promote translational activities. In contrast, phase separation of the purified FMRP, belonging to the same fragile X-related family as FXR1, leads to translational repression[68]. It has been proposed that different protein compositions and the states of post-translational modifications influence the partitioning of the translational machineries into distinct compartments and their activity outcomes[24]. Interestingly, a recent study based on single-molecule imaging showed that translation occurs in stress granules, previously thought to be translationally silent condensates, and mRNA partitioning into stress granules does not necessarily provoke a change in translational activities[69]. Thus, the functional effect of mRNA recruitment into phase-separated

condensates may be intricate, and represent convoluted consequences of various biochemical as well as physical properties of individual condensates.

In line with this view, our results shed light on how different material states can impact condensate functions. Biomolecular condensates are viscoelastic network fluids exhibiting time-dependent material properties[19]. The microrheological characterization of condensates of purified proteins or mixtures of proteins and RNAs showed that they behave as Maxwell fluids where elasticity dominates at short timescales but at longer timescales condensates exhibit more liquid-like behaviors[20,21]. The strengths of intermolecular interactions are intimately linked to the molecular mobility within condensates as well as thermodynamic stabilities and viscoelastic properties of

condensates[20,70]. In our study, through the rational design of condensate composition, we demonstrate that the solidification of condensates augments the sequestration-based translation inhibition of target mRNAs. This result suggests that in our system, the effect of translation inhibition relies on the ability of condensates to suppress the molecular dynamics of resident components. In general, the optimal material properties may be tailored for particular condensate functions, as suggested in tumor suppression[71] as well as bacterial growth[16]. Notably, RNA can tune the material properties of condensates in a manner dependent on their sequences or secondary structures[72,73]. Thus, it will be exciting to see whether an interesting interplay between mRNA localization, the material states of condensates, and translational activity may exist to fine-tune the translation status of individual mRNAs.

Our light-activatable system can be adopted to systematically build up the complexity of intracellular condensates in terms of composition and material property, thereby enabling the study of the causality between these macroscopic phase behaviors and condensate functions. For example, diverse protein domains can be fused to the core light-activatable module to confer desired compositions and functionalities. We anticipate that our strategy of tuning the material properties based on composition control can be similarly adapted in future studies. We envision that such optogenetic approaches will not only help elucidate the inner working of intracellular condensates, but also provide a useful platform for engineering synthetic condensates.

## Methods

### Animals and animal study approval
Animal care and experimental procedures were carried out in accordance with protocols approved by the Institutional Animal Care and Use Committee (IACUC) at Seoul National University (SNU) under license number SNU-191219-1-3. Wild-type C57BL/6 pups were obtained from Koatech (Gyeonggi-do, Republic of Korea). Both male and female pups were sacrificed for neuron culture.

### Generation of immortalized mouse embryonic fibroblast (MEF) cell lines
Embryonic day E14 mice were isolated from euthanized pregnant females. After the removal of uterine decidua, each embryo was transferred along with its yolk sac to a fresh dish with PBS. The Yolk sac was carefully removed, followed by removal of the fetus head and dark red tissues (heart and liver). Each prepared fetus was washed with PBS and transferred to a new culture dish. After addition of 0.25% trypsin-EDTA (Thermo Fisher Scientific), fetal tissue was minced thoroughly with two razor blades. The dish was incubated at 37 °C and 5% $CO_2$ in a humidified incubator for approximately 45 min. Following thorough digestion by pipetting up and down multiple times, MEF cells were plated onto 10 cm dish and were cultured in a growth medium consisting of Dulbecco's modified Eagle's medium (Gibco), 10% fetal bovine serum (Gibco), 10 U/mL Penicillin-Streptomycin (Gibco), and 0.5 μg/ml Fungizone (Gibco).

To immortalize MEF cells, the cells were passed onto a 6-well plate and transfected with a plasmid encoding SV40 Large T Antigen at the passage number of 1 or 2, using Lipofectamine 2000 (Invitrogen) or Fugene HD (Promega). After transfection, the cells were plated onto a 10 cm dish and treated with 30 μM of D, L-sulforaphane (Sigma). Surviving cells were selected and cultured in the growth medium.

### Cell culture
Lenti-X 293 T (Takara, 632180), immortalized MEF and U2OS (KCLB, 30096) cells were cultured in growth medium consisting of Dulbecco's modified Eagle's medium (GIBCO), 10% fetal bovine serum (GIBCO), and 10 U/mL Penicillin-Streptomycin (GIBCO), and incubated at 37 °C and 5% $CO_2$ in a humidified incubator.

### Primary hippocampal neuron culture
Hippocampi were dissected from both female and male postnatal day 1 (P1) pups from Actb-MBS-KI homozygous, and C57BL/6 wildtype (Koatech) mice. Hippocampi were digested with 0.25% Trypsin (Gibco) at 37 °C for 15 min. Trypsin was inactivated by incubating hippocampi in FBS containing plating medium (10% FBS (Gibco), 1x Glutamax (Gibco), 0.1 mg/ml Primocin (Invitrogen) in Neurobasal A medium (Gibco), followed by trituration and plating onto overnight poly-D-lysine-coated confocal dishes (SPL Life Science) in 5% $CO_2$, 37 °C incubator. After 4 h of plating, 2 ml of B27 medium (Neurobasal A medium (Gibco) supplemented with 1xB27 (Gibco), 1x GlutaMAX, (Gibco) and 0.1 mg/ml primocin (Invitrogen)) were added and neurons were grown in B27 medium until the experiment.

### Constructs
stdMCP-Cry2-mCh-FUS$_N$ was generated by inserting stdMCP, Cry2, mCherry, and FUS$_N$ (1–214) into a pHR-based vector. DNA fragments corresponding to stdMCP (Addgene, 98916) and Cry2 (Addgene, 10,1223) were amplified by PCR (Supplementary Data 1) and inserted into pHR-SFFV/mCherry/FUS$_N$ vector. If not specified otherwise, all fragment assemblies are performed using HiFi DNA Assembly Master Mix (NEB). miRFP670-optoFUS is identical to the optoFUS (Addgene, 10,1223) except for the fluorescent protein, swapped from mCherry to miRFP670. To create the TagBFP-DD-12xMS2PP7 construct, TagBFP (Addgene, 122151) and 12xMS2PP7(Addgene, 52984) were amplified by PCR. The destabilization domain (DD) was appended in-frame to the C-terminus of TagBFP by adding to the primer sequence. pUBC-NLS-HA-stdPCP-stdGFP was a gift from Robert Singer. pHR-hSyn-stdMCP-Cry2PHR-mCherry-FUS$_N$, which was used for expression in neurons, was generated by replacing the SFFV promoter with the hSyn promoter. stdMCP from this construct was deleted to clone pHR-hSyn-Cry2PHR-mCherry-FUS$_N$ which was used for the FISH experiment in neurons. The resulting constructs were fully sequenced to check the absence of unwanted products.

### Lentiviral transduction
Lentivirus was produced by cotransfecting the pHR transfer plasmid, pCMV-dR8.91, and pMD2.G (9:8:1, mass ratio) into 293 T Lenti-X (Takara Bio, 63,2180) cells grown to approximately 70% confluency in 6-well plates, using FuGENE HD Transfection Reagent (Promega) per manufacturer's protocol. A total of 3 mg plasmids and 9 μL of transfection reagent were delivered into each well. After 2 days, supernatant containing viral particles was harvested and filtered with 0.45 μm filter (Millex). Supernatant was immediately used for transduction or stored at − 80 °C in aliquots. Immortalized MEF or U2OS cells were grown to 10%–20% confluency in 12-well plates and 100–1000 mL of filtered viral supernatant was added to the cells. Cells infected were typically imaged no earlier than 72 h after infection.

### Single-molecule fluorescence in situ hybridization (smFISH)
Immortalized MEF cells transduced with optoMCP-FUS or optoFUS were activated for 20 min with blue light from a DIA lamp passing through a 452- nm band-pass filter (Edmund Optics, #86-351,). UBC-GFP and hSyn-optoMCP-FUS or hSyn-Cry2PHR-mCherry-FUS$_N$ cotransfected MBS-KI neurons (DIV 9-15) were activated for 20 min with blue light from a DIA lamp through a DAPI emission filter (Chroma, ET460/50 m). After 20 min of blue light activation, cells were fixed with 4% paraformaldehyde (Electron Microscopy Sciences) in 1x PBS-MC buffer (1x PBS supplemented with 1 mM $MgCl_2$ and 0.1 mM $CaCl_2$) on the microscope stage. Samples were continuously activated with blue light for 10 min immediately after the addition of fixative. Following 20 min incubation in the fixative, cells were washed three times with PBS-MC. Cells were then permeabilized with 0.1% Triton-X (Thermo Fisher Scientific) in 1x PBS-MC for 10 min, followed by washing twice with 1x PBS-MC and shacking for 10 min each. Before

hybridization with FISH probes (Supplementary Data 1), cells were preincubated in 10% formamide in 2X SSC for 10 min. Hybridization was carried out at 37 °C with 0.067 µM of FISH probes in hybridization buffer (2 mg/ml BSA (Roche, 10711454001), 10% Dextran Sulfate (Sigma-Aldrich), 10% formamide, 25 µg/ml single stranded DNA from salmon testes (Sigma-Aldrich, D7656), 25 µg/ml E. coli tRNA (Roche, 109 541) in 2X SSC) for at least 3 h up to overnight. After hybridization, samples were washed twice with 10% formamide in 2X SSC at 37 °C for 20 min each. Then, cells were subsequently washed with 2X SSC and PBS-MC for 10 min with shaking. After DAPI staining for 1 min, final washes with PBS-MC were performed more than twice for 10 min each. Samples were stored in PBS-MC at 4 °C before imaging. FISH probes are designed to bind to linker sequences between the stem loops. Sequences and fluorophores for all the probes are described in Supplementary Data 1.

### Stress granule immunocytochemistry experiment
U2OS cells expressing optoMCP-FUS and tagBFP-DD-12x(MBS-PBS) were globally activated with 452-nm band-pass filtered (#86-351, Edmund Optics) DIA lamp for 20 min. For stress treatment, oxidative stress was induced using 250 µM sodium arsenite (Sigma-Aldrich, S7400) for 30 min. Cells were washed with warm DPBS, and then fixed with Image-iT™ fixative solution (Invitrogen FB002) at room temperature for 5 min. Cells were then washed with DPBS twice and 0.5% PBS-T (PBS supplemented with 0.5% Triton X-100, Cayman chemical company, A35316) was treated for 5 min. After rinsing with 0.1% PBS-T, the cells were incubated with a blocking solution (0.1% PBS-T and 10% fetal bovine serum (FBS; HyClone, SV30207.02)) for 30 min. Next, cells are treated with mouse monoclonal anti-G3BP1 (Abcam, ab56574, 1:200 dilution) in blocking solution for 1 h, washed twice with 0.1% PBS-T, incubated with secondary antibodies specific to mouse IgG conjugated to Alexa 647 (Invitrogen, A-32728, 1:500 dilution) in blocking solution for 1 h, and washed two times with 0.1% PBS-T.

### OP-Puro incorporation
We used Click-iT™ OPP Alexa Fluor 647 Protein Synthesis Assay Kit (Thermo Fisher Scientific, C10458), and followed the manufacturer's instructions. Briefly, U2OS cells were globally activated with 452-nm band-pass filtered (#86-351, Edmund Optics) DIA lamp for 20 min. The cells were incubated with 20 µM Click-iT™ OPP (O-propargyl-puromycin) in a culture medium for 30 min at 37 °C, 5% $CO_2$ humidified microscope chamber. For the cycloheximide (CHX) control condition, cells were pretreated with 100 µg/mL CHX for 5 min, and then incubated with 20 µM OPP and 100 µg/mL CHX in culture media. After OPP labeling, cells were washed with warm DPBS, and then fixed with Image-iT™ fixative solution (Invitrogen FB002) at room temperature for 5 min. Cells were then washed with DPBS two times.

### Puro-PLA
Before the addition of puromycin, immortalized MEF cells were globally activated with 452-nm band-pass filtered (#86-351, Edmund Optics) DIA lamp for 20 min. For puromycin labeling, cells were incubated with 3 µM puromycin (Tocris) in culture medium for 5 min at 37 °C, 5% $CO_2$ humidified microscope chamber. For the anisomycin control condition, cells were first incubated in 40 µM anisomycin for 20 min without blue light activation, and then puromycin labeling was conducted in the presence of 40 µM anisomycin and 3 µM puromycin for 5 min. After puromycin labeling, cells were quickly washed twice with warm 1X PBS-MC and then fixed with 4% PFA in 1X PBS-MC. After the fixation and washing, cells were permeabilized with 0.5% Triton X-100 in PBS-MC for 15 min followed by 2 times washing with PBS-MC. Cells were incubated with blocking buffer (4% goat serum in 1X PBS-MC) for 1 h at 37 °C. Cells were incubated with Anti-Puro (1:3000, EQ0001, Kerafast) and Anti-Beta-actin (1:1000, ab8227, Abcam) diluted in the blocking buffer, at 37 °C humidified chamber overnight. For the

following rolling circle amplification, we used Duolink In Situ PLA Probe Anti-Rabbit Plus (DUO92002, Sigma-Aldrich), Anti-Mouse Minus (DUO92004, Sigma-Aldrich), and Detection Reagents Green (DUO92014, Sigma Aldrich) following manufacturer's instruction as described in the previous studies[49,56]. Wash buffers 'A' and 'B' were equilibrated to room temperature before use. After incubation with antibodies, cells were washed two times with buffer 'A' for 5 min. After the vortexing of PLUS and MINUS PLA probes, each probe was diluted 1:5 in Duolink Antibody Diluent. PLA probe solution was applied to cells and cells were incubated in a pre-heated humidity chamber for 1 h at 37 °C. Following the incubation, cells were washed with buffer 'A' for 5 min twice. 5X Duolink Ligation buffer was diluted 1:5 in high-purity water. After the wash, Ligase was added to 1X Ligation buffer at a 1:40 dilution. Ligation solution was added to cells and cells were incubated in a pre-heated humidity chamber for 30 min at 37 °C. 5X Amplification buffer was diluted 1:5 in high-purity water. Following incubation, cells were washed with buffer 'A' for 5 min twice. After washing, Polymerase was added to 1X Amplification buffer at a 1:80 dilution. Amplification solution was added to cells and cells were incubated in a pre-heated humidity chamber for 100 min at 37 °C. After the incubation, cells were washed twice with buffer 'B' for 10 min. Extra wash with 0.01X buffer 'B' was performed for 1 min. Cells were imaged by microscope without mounting. For a more detailed description, please refer to Duolink PLA Fluorescence Protocol (Sigma-Aldrich).

### Chemical LTP (cLTP)
Homozygous MBS-KI or wildtype neurons were co-transfected with iRFP filler and hSyn-optoMCP-FUS$_N$ 36–48 h before cLTP experiments with lipofectamine 2000 (Invitrogen). 6 µl of lipofectamine 2000 were diluted in 75 µl OptiMEM, followed by incubation at room temperature for 5 min. 1.5 µg of iRFP and 1.5 µg of hSyn-optoMCP-FUS$_N$ plasmids were diluted in 75 µl OptiMEM and were mixed well. 75 µl of plasmid solution were added to the lipofectamine solution dropwise and again mixed by pipetting. After 10 min incubation at room temperature, 150 µl of mixture was added to the center of the cell-residing confocal dish. DIV 18–21 hippocampal neurons were incubated in extracellular solution (150 mM NaCl, 2 mM $CaCl_2$, 5 mM KCl, 10 mM HEPES, 30 mM glucose, 1.5 µM TTX, 20 µM Bicuculline, and 6 µM Strychnine) for 40–60 min. Baseline images of spines were imaged every 3 min, three times, for 9 min before stimulation. Neurons were stimulated with an extracellular solution containing 200 µM glycine for 3 min. After stimulation, neurons were washed 3 times with extracellular solution, and were imaged every 5 min, 36 times. Blue light activation was performed by shining a DIA lamp through a DAPI emission filter (ET460/50 m, Chroma) for 20 min before taking baseline images of the spine and during 3 min of glycine stimulation. During spine imaging of baseline and after cLTP treatment, z-stack images of GFP and iRFP channels were taken.

### Microscopy
All confocal images are taken using 60X oil immersion objective (NA 1.4) on a Nikon A1 laser scanning confocal microscope. An imaging chamber is maintained at 37 °C and 5% $CO_2$. For live cell imaging, cells are plated on the fibronectin (Millipore Sigma) coated 4-chamber 35-mm glass-bottom dishes (Cellvis) and grown typically overnight. For single-cell activation, cells are imaged with a 488-nm laser every 15 s. For global activation, cells are typically imaged with either a 488-nm laser or 452-nm band-pass filtered (#86-351, Edmund Optics) DIA lamp. For local activation, a region of interest (ROI) is defined to guide the area to be scanned with a blue laser. Fluorescence recovery after photobleaching (FRAP) is performed similarly using ROI after 15 min of activation with 15-s intervals.

smFISH images for immortalized MEF cells and neuronal cells were taken using Olympus 150X oil immersion objective (NA 1.45) on Olympus IX73 inverted microscope equipped with two iXon Ultra 897

EMCCD cameras (Andor), and MS-2000XYZ automated stage (ASI). In immortalized MEF FISH experiments, a 488- nm diode laser (Cobolt) was used to excite both β-actin mRNA labeled with FAM FISH probes and stdMCP-Cry2PHR-mCherry-FUS$_N$ condensates. Emission from both fluorophores was separated by LED-FITC-A-000 filter set (Semrock) and imaged with two EMCCD cameras respectively. For simultaneous detection of MS2-tagged β-actin mRNA, stdMCP-Cry2PHR-mCherry-FUS$_N$ condensates, and GAPDH mRNA, a 488-nm diode laser (Cobolt) was used to excite β-actin mRNA labeled with FAM FISH probes. Emission light was reflected by dichroic mirror FF495-Di03-25 × 36 (Semrock) and passed through emission filter FF01-525/45-25 (Semrock). stdMCP-Cry2PHR-mCherry-FUS$_n$ condensates were excited with 561- nm diode laser (Cobolt). Emission light was reflected by the dichroic of the ET-Cy5 49006 filter (Chroma). Quasar 670 FISH probes labeled GAPDH mRNAs were excited by 647-nm diode laser (Cobolt), and emission light was passed through dichroic of ET-Cy5 49006 filter (Chroma), and extra ET700/50 m (Chroma) bandpass filter. The same imaging set-up was used for neuron FISH experiments to visualize stdMCP-Cry2PHR-mCherry-FUS$_N$ or Cry2PHR-mCherry-FUS$_N$ condensates and MS2-tagged β-actin mRNAs labeled with Quasar 670 FISH probes.

smFISH images for U2OS cells were taken using Nikon 100X oil immersion objective (NA 1.49) on Nikon Eclipse Ti-E inverted wide-field microscope equipped with an iXon Ultra 897 EMCCD camera (Andor), and a XY-scanning module with NanoDrive PiezoZ (MCL). The LU-N4 laser unit equipped with 405 nm, 488 nm, 561 nm, 640 nm lasers (Nikon) was used to excite mRNA labeled with Alexa-488 FISH probes, optoMCP-FUS condensates and miRFP-optoFUS condensates. Emission was filtered by C-FL-C TRITC, C-FL-C-FITC (Nikon), or 49009 ET - Cy5 (Chroma) filter cubes.

Live cell single-molecule imaging of MS2-tagged β-actin mRNAs through stdMCP-Cry2PHR-mCherry-FUS$_N$ in immortalized MEF cells was conducted using Olympus IX83 inverted wide-field microscope. The microscope is equipped with Olympus 150X oil immersion objective (NA 1.45), iXon Life 888 EMCCD camera (Andor), TANGO 3 desktop stage controller (Marzhauser), SOLA SE white light-emitting diode (Lumencor), and ET-39010 filter set (Chroma). The Stage-top Incubator System TC (Live Cell Instruments) maintained the temperature of the humidified imaging chamber at 37 °C and CO$_2$ concentration to 5%. Live-cell images of dendritic spines after chemical LTP treatment were taken using the same setup except for the 49022-ET-Cy5.5 filter (Chroma) used for imaging the iRFP filled dendritic spines. To orthogonally image c-FOS-12x(MBS-PBS) mRNA and optoMCP-FUS condensates, 488- nm diode laser (Cobolt) was used, and emission light passed same filter sets used for imaging stdMCP-Cry2PHR-mCherry-FUS$_N$ condensates and FAM-FISH probes. For tracking single mRNAs in live cells, the MEF cell is imaged every 50 ms after 30 s of blue light activation. Since Cry2 is highly sensitive to 488-nm laser, we were very cautious not to expose 488- nm laser to cells prior to imaging.

### Puncta image analysis
To measure mCherry and/or miRFP signals in light-activated condensates, images were blurred and background-subtracted in ImageJ using the Subtract Background and the Gaussian Blur plugins. The images went through thresholding, and regions of puncta were set by the ROI manager in the ImageJ using the Analyze Particles plugin.

### Image analysis for live-cell single-molecule mRNA imaging
Positional shifts between two EMCCDs were corrected with custom-built MATLAB codes and images of fluorescent beads. To quantify the temporal evolution of mRNA recruitment, photobleaching was first corrected with the Bleach Correction ImageJ plugin. Then, mCherry channel images were band-passed and then thresholded to determine the boundary of the optoMCP-FUS condensates. stdPCP-

stdGFP signals within the boundary of the optoMCP-FUS condensates were measured to quantify the integrated GFP intensity in condensates. For tracking of mRNAs and condensates in high-temporal resolution, images were analyzed with custom-built MATLAB scripts[74].

### Spine image analysis
Regions of spines were set by ROI manager in the ImageJ. ROI was drawn manually to include spines in every *XYZ* directions. Before subtracting the backgrounds, raw images were divided by flat field image generated by random 400 images with the BaSiC plugin in the ImageJ[75]. Then, the Correct 3d Drift and the Bleach Correction plugins were used to correct the images from drift and photobleaching. To subtract the backgrounds of images, the Subtract Background plugin and the Phansalkar method in the Auto Local Threshold plugin were used. Volumes of spines were measured by summing the slices from the z-stack.

### FISH image analysis
For immortalized MEF FISH experiments, two-channel images were first divided by respective flat-field images generated from the BaSiC imageJ plugin in order to correct the non-uniform illumination. Next, positional shifts between two EMCCDs were corrected with custom MATLAB codes and fluorescent-bead images. Binary masks of cytoplasm and nucleus were generated by the ROI manager in ImageJ. Light-induced condensates (mCherry) and MBS-tagged β-actin mRNAs (FAM) were localized at the subpixel resolution with TrackNTrace[76] from z-stack images covering the whole cells with the z-step of 0.5 μm. Puncta were tracked through z-images and the brightest z-planes were designated as the z-position of puncta. The closest distances between pairs of condensates and mRNA puncta were calculated. β-actin mRNA puncta within the distance of less than 330 nm (~3 pixels) to the condensates in XY and within two z-steps were considered to be recruited to the condensates. To quantify the fraction of recruited mRNAs in condensates, amplitudes of mRNA puncta were used. For this purpose, images of MEF smFISH samples were processed with ImageJ built-in Gaussian Blur plugin with sigma (radius) of 1. Then, the amplitude of each mRNA punctum was calculated by subtracting the minimum pixel intensity from the maximum value within 81 pixels (9 × 9) surrounding the localized punctum. For each cell, amplitude-based mRNA recruitment was calculated by dividing the sum of amplitudes of recruited mRNA puncta by the sum of amplitudes of all mRNA puncta within the cell.

For neuron FISH experiments, single focused z-plane was analyzed. Binary masks of dendrites were generated by the ROI manager in ImageJ, and TrackNTrace was used to detect positions of light-activated condensates (mCherry) and MBS-tagged β-actin mRNAs (FAM). β-actin mRNAs with the distance of less than 330 nm (~3 pixels) were considered to be recruited to the condensates. For each dendrite, amplitude-based mRNA recruitment was calculated by dividing the sum of 2D-Gaussian fitted amplitudes of recruited β-actin mRNAs by the sum of 2D-Gaussian fitted amplitudes of all β-actin mRNAs present in the dendrite. For U2OS FISH experiments, images were analyzed using custom-built MATLAB scripts (modified from an open-source code[74]). First, two-channel images were processed to correct the non-uniform illumination using dilute FAM dye samples. Next, light-activated condensates (mCherry) were localized using the MATLAB script from z-stack images covering the whole cell with the z-step of 0.5 μm. Using band-pass filtered FISH images, mRNA recruitment was quantified by dividing the sum of mRNA signals within the distance of less than 750 nm (~7 pixels) to the localized condensates in XY and within two z-steps by the sum of all mRNA signals within each cell. Integrated mRNA puncta intensities were calculated by the sum of pixel intensities of all localized mRNA puncta within individual cells.

## OP-Puro image analysis

For OP-Puro incorporation assay[77,78], z-stack cell images were max-Z projected. Nuclei were detected using NLS-stdPCP-stdGFP signals. GFP signals were blurred and background-subtracted in ImageJ using the Smooth and the Subtract Background plugins, respectively. Binary images were generated using the Threshold plugin, then nuclei were detected using the Watershed and the Analyze Particles plugins. Detected masks were manually inspected. For each cell, mean OPP intensities within the mask were measured across all z slices, and the maximum value was chosen as the corresponding OPP intensity of the given cell.

## Puro-PLA image analysis

For immortalized MEF Puro-PLA experiments, positions of Puro-PLA puncta were detected by TrackNTrace from z-stack images covering the whole cell with the z-step of 0.5 μm. Binary masks of each cell were generated by the ROI manager in ImageJ. For each cell, Puro-PLA puncta number were calculated and plotted.

## Mean squared displacement (MSD) analysis

To measure MSD, photobleaching was corrected with the ImageJ plugin, and then images were band-passed to remove noise. Puncta were identified and tracked at sub-pixel resolution using custom-built MATLAB scripts. The diffusion constant was determined by fitting the $2^{nd}$, $3^{rd}$, and $4^{th}$ data points of the MSD to a linear line[79].

## Statistics and reproducibility

Results are presented as mean ± standard deviation except for Fig. 5F–H, Fig. S2B, and S3C, where the center line or bar is the mean and error bars are standard error of the mean. Significant differences were analyzed with two-tailed Student's $t$ test using Microsoft Excel software. All micrographs in this study are representative images of experiments carried out with at least three repetitions.

## Reporting summary

Further information on research design is available in the Nature Portfolio Reporting Summary linked to this article.

## Data availability

The data supporting the findings of this study are available from the corresponding authors upon request. Source data are provided in this paper.

## Code availability

The source code for detecting and tracking particles or condensates is available at GitHub (https://github.com/MinLeeKR/optoMCP-FUS).

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

## Acknowledgements

This work was supported by the National Research Foundation of Korea (2019R1C1C1006477, 2022R1A5A102641311, RS-2023-00211612, RS-2023-00260454 to Y.S., 2020R1A2C2007285 to H.Y.P.). Y.S. was also supported by the Creative-Pioneering Researchers Program of Seoul National University, and the Samsung Science and Technology Foundation under Project Number SSTF-BA1901-12. H.Y.P. was also supported by the Samsung Science and Technology Foundation under Project Number SSTF-BA1602-11. Y.S. also acknowledges administrative support from SNU-IAMD.

## Author contributions

Conceptualization, Y.S., H.Y.P.; Methodology, M.L., H.C.M., H.J., and D.W.K.; Investigation & Analysis, M.L., H.C.M., and H.J.; Writing-Original Draft, Y.S.; Writing-Review & Editing, M.L., H.C.M., H.J., D.W.K., H.Y.P., and Y.S.; Funding acquisition & Supervision, Y.S. and H.Y.P.

## Competing interests

The authors declare no competing interests.
