## [Peer Review File · Nature Communications]

Optogenetic control of mRNA condensation reveals an intimate link between condensate material properties and functionsREVIEWER COMMENTS

Reviewer #1 (Remarks to the Author):

Lee et al. use the well-described system of optogenetic phase separating proteins, based in this case on FUS intrinsically disordered region (IDR), to test the hypothesis that mRNA translation depends on the degree to which a condensate can sequester the mRNA, which in turn, depends on the material properties of the condensate. To achieve this they fuse FUS IDR to tandem MS2 coat protein (stdMCP) which can recruit target mRNA species containing bacteriophage MS2 binding sites (MBS), of which an array was integrated into mRNA-encoding sequences of the actin gene expressed from a plasmid. The co-expressed these two components with optoFUS, the blue light-activated phase separating protein. Importantly, at different expression levels, optoFUS can produce liquid to gel-like properties. Using single molecule observations they take advantage of the broad variations in abundances of the proteins in different transfected MEF cells to construct a phase diagram in which three categories of condensates formed, 1 those in which the mRNA is not partitioned, is partitioned but show dynamics and FRAP typical of a liquid condensate and those which show behaviors of a gel, as predicted. Importantly, translation of the RNA was repressed along the continuum of optoFUS concentration and therefore gelness. The authors illustrated how synaptic spine dynamics could be modulated by putting expression of actin under control of optoFUS.

This manuscript is well written and the experiments performed. With rigor and appropriate statistical analyses. Whether the authors observations have biological relevance is not known, however the technology itself may be of interest to those interested in modulating gene expression in a highly localized way. The work, however, may explain differential effects of mRNA translation by the protein Whi3 in filamentous yeast (Zhang, H. et al. RNA Controls PolyQ Protein Phase Transitions. *Molecular Cell* 60, 220-230 (2015). <https://doi.org:10.1016/j.molcel.2015.09.017>). In that study, the authors showed that the Whi3 condensate sequestered and repressed translation of one mRNA for the cyclin CLN3, but did not repress that of another mRNA, BNI1. The authors attributed these differences to the different viscosities of the Whi3-mRNA condensates, but their results are ambiguous, potentially arising from differences in elasticity that of the condensates. The manuscript would benefit from some discussion of this paper and implications to other cases of differential repression of translation by other natural condensates.

Reviewer #2 (Remarks to the Author):

In the study by Lee et al. the authors investigate the effect of mRNA sequestration into biomolecular condensates on translational output. To this end, they formed synthetic condensates using an optogenetic approach to aggregate the IDR domain of FUS, as an approach to control the RNA-binding proteins that associate with target mRNAs via MS2 or PP7 loops. They demonstrate that condensation of mRNAs leads to rapid translational repression, which is stronger in cells expressing higher concentrations of the phase-separating component. They note the highly transient nature of interactions between individual mRNAs and condensates, as observed in single-molecule tracking experiments. By expressing different constructs, the condensates could be sufficiently manipulated, and could have their fluid liquid state altered into a more rigid, solid-like granule. This then intensified the translational repression. Lastly, they show how by using their optogenetic system it is possible to investigate whether condensate-based translation inhibition can further modulate downstream cellular functions. They show that the formation of condensates sequesters specific endogenous mRNA species in neurons, which can modulate localized cellular activities such as the spine enlargement associated with synaptic plasticity.

Altogether the insights of this study are limited. Aside from the nice technical developments, the

authors basically only show a dampening in translation from mRNAs sequestered in the granules, but this is not surprising or novel considering that studies on cytoplasmic granules that have mRNAs within, have shown that these are non-translating mRNAs. In that respect, it is unclear whether the properties of the condensates that are formed here have any correlation to naturally occurring phase-separated granules that contain mRNAs. Additionally, although completely logical, the notion that higher levels of expression yield higher translational repression is patently obvious. There is more protein that is binding and sequestering more mRNA that cannot engage with the polysome and active translation. This again does not seem like a novel insight.

General remarks:

* Using present tense in the Results section is quite unusual causing some sentences to sound strange. Past tense should be used.

* Combining red and green fluorescent images has been out of practice for many years. See: <https://www.ascb.org/science-news/how-to-make-scientific-figures-accessible-to-readers-with-color-blindness/>

Major issues:

* Fig. 1J – the FRAP experiment is performed on a scale of minutes. The recovery is approximately 50%. This means that the condensates are actually quite stable structures and that ~50% of a condensate's components are stuck within and not moving out for a long time. This indicates that this structure is quite rigid whereas typical protein components of cellular condensates/granules have much faster recovery times, on the range of seconds. In other words, naturally occurring phase-separated structures have ongoing influx and outflux of components which is very different from the condensates formed here. I therefore do not think they are like liquid droplets as stated and do not think they show a significant degree of fluorescence recovery. This is actually confirmed later in figure 4.

* It seems that condensates behave differently with or without RNA (Fig. 1H D). What would the explanation for this be? Could the mRNA somehow interfere with the condensation because in Fig. 1D there are single mRNAs without the MCP signal that appear the same size as 1 red puncta with RNA, while without RNA (Fig. 1D optoFUS) the mCherry dots look 2-3 fold larger. Could try activating the optoMCP-FUS longer to obtain "mature" condensates without MCP.

* Fig. 2B – line 162 – "we were able to capture the dynamics of the reporter mRNA recruitment into the condensates" – why is recruitment the term to describe what is seen. Rather, the mRNAs are there to begin with and the proteins that are bound to them aggregate and pull them along, or simply form around them. I think this experiment shows the formation of condensates that contain mRNAs within.

* Fig. S2B – faster diffusion – but there are no diffusion measurements. One would need a diffusion coefficient to say something about diffusion properties. The figure shows that the mRNAs are freer to diffuse than the condensates. Looking at the movies in which the condensates don't seem to move very much and taking into account the FRAP interpretation above of a rigid structure, these structures are not really comparable. Also, the sentence "even though their sizes appear to be similar in the diffraction-limited images" has no real significance – one cannot infer such interpretation from them looking similar under a microscope. Finally, a control FRAP experiment of the condensate without mRNA should have been performed to understand the effect of the RNA on the biophysical properties of the condensate.

* Fig. 2E,F- Line 171 – "Strikingly" – what is so striking about them meeting and separating. Also, how frequent are these encounters – do all condensates acquire the mRNAs that way, because beforehand the authors wrote that the mRNAs act as oligomerization scaffolds, which would mean that they nucleate from the mRNA or together.

Lines 168-182: What do the authors suspect is the mechanism as to why some of the mRNAs tether tightly and others just graze the surface of the condensates. The authors suggest that the variance is due to translation capacity of the transcripts, but what would make some transcripts primed for translation and not others? Although many studies have documented that different mRNAs have different associations to various condensates, it appears there is significant variation among these identical constructs. One thing this reader was left wondering was what is the relationship between

these condensates and overall translation of the cell. Does it affect one of the two canonical phosphorylation pathways that inhibit translation and yield stress granule condensates? (Should be mentioned that it has been shown in recent years that RNA in stress granules can be translationally active.) Although in figure 3 translational capacity was addressed, it seems relevant to understand general translation occurring at global cellular level, rather than the one transcript interacting directly with the condensate. These condensates form and disperse rapidly, but figure 3 demonstrates that the light can be administered for many hours. This questions how a huge number of large condensates affects translation, and not just one transcript.

Also, in Fig. 2E it appears that the RNA can be bound or unbound or even associated (leaving and coming back), it would be interesting to measure a diameter for "associated" RNA and see if shifts between unbound-bound-associated can be measured over time. Namely can the 'saturation level' (line 167) change.

* Fig. 3: More protein yields more aggregates which increases translation inhibition. This is then shown in the data in Fig. 3F under light activation. But a control is missing, of translation levels when no light is administered. What effects do high levels of this construct have on the transcripts, prior to formation of condensates. Otherwise, although it seems logical that the condensates directly impact on mRNA functionality and not just more RNA binding, this has to be shown. This point is also missed in the scheme in Fig. 3A, where it shows red condensates forming after addition of blue light, but no red whatsoever in the cells in the untreated state. This is of course not the case, and the FUS construct is free in the cell.

* The story shifts to an unclear path around figure 4. The analysis could be interesting as a physics examination but doesn't really add much to the story. Basically, the translation reduction was concluded without all this.

* Fig. S4B- The purpose of the experiment of puro-PLA should be explained in more detail. Which puncta are being considered PLA-positive? The images are small and its unclear what is being counted. Does the b-actin antibody bind to an epitope at the very N-terminus of the protein? Also, what is the significance of the experiment. Probably more controls should be shown. The conclusion is that the translation inhibition occurs within 20 minutes, but would there be reason to think otherwise?

Minor:

* The authors show in Fig. 1 the optoFUS without MCP, however, a control of the construct without the FUS domain is necessary. The authors claim based on previous research that the non-RNA binding IDR of FUS causes the condensation after blue light, but what will the MCP-Cry2-mCherry do under blue light?

* Fig. 1D: It will be preferable to show the entire cell together with the enlargement. This will give a better indication on what we are seeing. Also, are the condensates in the cytoplasm only, or also in the nucleus. In case there are also nuclear condensates, then in the b-actin MEF cells where active genes are observed in the nucleus, are these also associated with the condensates?

Also, can barely see mRNA FISH signal in the OptoFUS section. Suggest to indicate with arrows what is mRNA and what is not.

* Fig. S1B: Should add the entire cell image with the enlargement.

* The authors addressed the lack of visible transcripts bound to the MCP-FUS, and this was thought to be owed to the high levels of expression. But actually in the movies provided (414494_0_video_7354989_rq623n) the expression level is quite low, and the condensates appear rapidly after the FUS begins to aggregate. In this case, where are the transcripts before blue light exposure. In theory they are bound to the coat protein before aggregation begins, and there is no issue of overexpression. The authors mention the tight binding of MCP to the MS2 loops.

* Were other wavelengths of light tested, to see if condensates will form upon exposure to other wavelengths? Also, throughout the manuscript it seems that different wavelengths (wide filters) are used to excite the Cry2 system. This implies excitation around 488nm, the same as the GFP channel. Wouldn't the overlap of wavelengths cause a "blue" light excitation? For instance, in line 163 "Prior to blue light exposure, individual mRNAs are visible in the GFP channel..." wouldn't the GFP imaging cause the condensation of the MCP proteins and thus may affect results? In prior work of the authors they showed that different wavelengths can activate the Cry2 system. Suggest to add a little more detail about this, while mentioning why different times are used for activation in every experiment.

- * Because there is such a dependence on mCherry-FUS expression in order to yield condensate formation, there is a question of how many protein/RNA complexes are found in this granule. From Fig. 1 we see that unbound mRNA in the cells expressing just optoFUS have a certain intensity (green). By measuring this intensity, perhaps it is possible to calculate how many mRNPs are found at the condensate in the opto-MCP-FUS, and then also get an answer as to how many of these RNA/protein complexes are necessary for formation (because there is a direct relation between the protein and RNA) rather than the somewhat vague term 'high expression'. It might not be too difficult to calculate the number of mRNAs present, and thus imply the number of FUS in each condensate, especially given that these condensates seem uniform in size.
- * Line 158/161 - "proper levels" "suitable levels" - I suppose the authors mean the ideal levels of expression that yield clear signal over the background.
- * Line 169 - Explain 20 Hz. Please add some detail at least in the materials section.
- * Line 189 - why 12 hours of activation? Is 20 min not enough like anisomycin (which is not mentioned in the materials aside of PLA context). Wouldn't 12 hours of blue light be toxic to the cells? Should show cell survival analysis to see that the cells are healthy.
- * Fig. 4G - looks like in the small squares that group 2 regains its signal faster and stronger than group 1, not like the graph below - why? Please add to every FRAP experiment the time of blue light activation, at least in the materials sections. Would a condensate immobile fraction change from using different times to activate? Please show FRAP at different times of light (5min/20min/12hours). Number of cells measured in FRAP experiments is low.
- Fig. 4 - would the RNA accumulate differently in every group? Please check with smFISH the different mRNA sequestration to each group, it would strengthen your point.
- * Fig. S4B - It is known that expression levels of a certain RNA-binding protein can itself change granule abundance mRNA dynamics. This is especially relevant when in Fig. S4D it is shown that the levels of mCherry alone change PLA puncta. Therefore, the lowering of PLA puncta after condensate formation in Fig. S4B may be affected by the amount of FUS expressed before light activation. Therefore, an image of the cell pre-light exposure should be shown in order to confirm that the amount of FUS in each cell that are now being compared are identical.
- * Movie S4- add another movie of the stabilized interaction
- Materials - smFISH- add the probes sequences, and add to where the probe binds- the MS2 loops or the mRNA exons.
- * Plugin are used to reduce noise. Explain if these could affect the analysis. Line 525: what's the plugin's name? add "Bleach correction" plugin if it's that.
- * For making this study more insightful, one could tag a few different proteins such as translation initiation factors together with the Cry2 system and see if there is any change in translation.

Reviewer #3 (Remarks to the Author):

the manuscript by Lee et al, describes and optogenetic approach to initiate phase transition of a mRNA by blue light induced RNA-protein complex formation. The approach described uses the well studied MS2 coat protein (and variants thereof) for binding to mRNA that were modified with cognate aptamer domains. To gain light sensitivity, a cry2 system for light-triggered homo dimerisation and multimerisation was applied.

The authors showed that upon multimerisation, hence phase transition of the mRNA in cells the translational activity of respective RNAs was diminished. This feature is somehow expected, as ribosomal access to mRNAs will be inhibited in condensates, at least sterically. The authors compare translational activity of condensed mRNA vs non-condensated mRNA. The authors extended their

approach to endogenous proteins/mRNA (actin). However, I am wondering what is the impact of the MS2 coat protein RNA assembly on translational efficacy per se? Other than that, I think the quality of the MS is very high and the findings are of broad interest. Therefore, recommend acceptance.

Response to Reviewers :

We thank the reviewers for their constructive comments and have revised the manuscript according to their suggestions. We provide a detailed point-by-point response to each of the reviewers' concerns below (our responses are in blue text):

REVIEWER COMMENTS

Reviewer #1 (Remarks to the Author):

Lee et al. use the well-described system of optogenetic phase separating proteins, based in this case on FUS intrinsically disordered region (IDR), to test the hypothesis that mRNA translation depends on the degree to which a condensate can sequester the mRNA, which in turn, depends on the material properties of the condensate. To achieve this they fuse FUS IDR to tandem MS2 coat protein (stdMCP) which can recruit target mRNA species containing bacteriophage MS2 binding sites (MBS), of which an array was integrated into mRNA-encoding sequences of the actin gene expressed from a plasmid. The co-expressed these two components with optoFUS, the blue light-activated phase separating protein. Importantly, at different expression levels, optoFUS can produce liquid to gel-like properties. Using single molecule observations they take advantage of the broad variations in abundances of the proteins in different transfected MEF cells to construct a phase diagram in which three categories of condensates formed, 1 those in which the mRNA is not partitioned, is partitioned but show dynamics and FRAP typical of a liquid condensate and those which show behaviors of a gel, as predicted. Importantly, translation of the RNA was repressed along the continuum of optoFUS concentration and therefore gelness. The authors illustrated how synaptic spine dynamics could be modulated by putting expression of actin under control of optoFUS.

This manuscript is well written and the experiments performed. With rigor and appropriate statistical analyses. Whether the authors observations have biological relevance is not known, however the technology itself may be of interest to those interested in modulating gene expression in a highly localized way. The work, however, may explain differential effects of mRNA translation by the protein Whi3 in filamentous yeast (Zhang, H. et al. RNA Controls PolyQ Protein Phase Transitions. *Molecular Cell* 60, 220-230 (2015). <https://doi.org/10.1016/j.molcel.2015.09.017>). In that study, the authors showed that the Whi3 condensate sequestered and repressed translation of one mRNA for the cyclin CLN3, but did not repress that of another mRNA, BNI1. The authors attributed these differences to the different viscosities of the Whi3-mRNA condensates, but their results are ambiguous, potentially arising from differences in elasticity that of the condensates. The manuscript would benefit from some discussion of this paper and implications to other cases of differential repression of translation by other natural condensates.

We thank the reviewer for positive comments. In the revised manuscript, we cited the Whi3 paper and also further discussed biological relevance of our study in multiple places (all changes in the manuscript are highlighted in blue text). As a central component of biomolecular condensates, RNA has received huge attention from the birth of the field. Previous works mainly identified key structural roles of RNA in modulating the phase behaviors of RNA-binding proteins. However, the effect of condensation on the translation activity of recruited mRNAs, especially with respect to changes in the material properties of condensates, has been unclear. Although condensates such as stress granules and germ granules are often associated with repressed mRNAs, recent studies showed several examples of condensates

with translationally active mRNAs, suggesting that the functional effects of mRNA condensation are more intricate than conventional views. In this study, we showed that sequestering mRNA into the biomolecular condensates with reduced internal dynamics has strong repressive effects on translation activities. We feel that our optogenetic system will be highly useful to build up the complexity of biomolecular condensates within living cells and to further dissect the relation between diverse condensate features and functional outputs, as we took initiative in this study.

Reviewer #2 (Remarks to the Author):

In the study by Lee et al. the authors investigate the effect of mRNA sequestration into biomolecular condensates on translational output. To this end, they formed synthetic condensates using an optogenetic approach to aggregate the IDR domain of FUS, as an approach to control the RNA-binding proteins that associate with target mRNAs via MS2 or PP7 loops. They demonstrate that condensation of mRNAs leads to rapid translational repression, which is stronger in cells expressing higher concentrations of the phase-separating component. They note the highly transient nature of interactions between individual mRNAs and condensates, as observed in single-molecule tracking experiments. By expressing different constructs, the condensates could be sufficiently manipulated, and could have their fluid liquid state altered into a more rigid, solid-like granule. This then intensified the translational repression. Lastly, they show how by using their optogenetic system it is possible to investigate whether condensate-based translation inhibition can further modulate downstream cellular functions. They show that the formation of condensates sequesters specific endogenous mRNA species in neurons, which can modulate localized cellular activities such as the spine enlargement associated with synaptic plasticity.

Altogether the insights of this study are limited. Aside from the nice technical developments, the authors basically only show a dampening in translation from mRNAs sequestered in the granules, but this is not surprising or novel considering that studies on cytoplasmic granules that have mRNAs within, have shown that these are non-translating mRNAs. In that respect, it is unclear whether the properties of the condensates that are formed here have any correlation to naturally occurring phase-separated granules that contain mRNAs. Additionally, although completely logical, the notion that higher levels of expression yield higher translational repression is patently obvious. There is more protein that is binding and sequestering more mRNA that cannot engage with the polysome and active translation. This again does not seem like a novel insight.

We thank the reviewer for providing constructive comments. We have updated our manuscript accordingly as elaborated below. Regarding the novelty of our work, we would respectively disagree with the assertions that “the authors basically only show a dampening in translation from mRNAs sequestered in the granules”, and that mRNAs within cytoplasmic granules are non-translating. As stated in the response to the reviewer 1, the functional effects of mRNA recruitment to condensates are still largely unclear, with several opposing examples showing both cases of active and inactive translation depending on the types of biomolecular condensates involved (PMID: 32783880, 35951695, 31439799). Our work goes well beyond

the simple demonstration of sequestration-based translation repression, by 1) Visualizing interactions between individual mRNA molecules and condensates (Fig. 2), 2) Probing the effects of the material properties of condensates on mRNA translation (Fig. 4), and 3) Studying the functional effects of localized translation inhibition in the neural plasticity (Fig. 5). Our paper illustrates how optogenetic approaches can be used to build up the complexity of biomolecular condensates in a systematic manner within living cells.

General remarks:

* Using present tense in the Results section is quite unusual causing some sentences to sound strange. Past tense should be used.

* Combining red and green fluorescent images has been out of practice for many years. See:

<https://www.ascb.org/science-news/how-to-make-scientific-figures-accessible-to-readers-with-color-blindness/>

We thank the reviewer for these comments, and have updated our manuscript accordingly.

Major issues:

* Fig. 1J – the FRAP experiment is performed on a scale of minutes. The recovery is approximately 50%. This means that the condensates are actually quite stable structures and that ~50% of a condensate's components are stuck within and not moving out for a long time. This indicates that this structure is quite rigid whereas typical protein components of cellular condensates/granules have much faster recovery times, on the range of seconds. In other words, naturally occurring phase-separated structures have ongoing influx and outflux of components which is very different from the condensates formed here. I therefore do not think they are like liquid droplets as stated and do not think they show a significant degree of fluorescence recovery. This is actually confirmed later in figure 4.

We first note that generally, biomolecular condensates are viscoelastic, exhibiting time-dependent material properties ranging from liquid-like to solid-like states (PMID: 33303613, 35675815). In other words, condensates behave differently depending on the time scale of external perturbations. In the revised manuscript, we have updated the figure showing condensate fusion (Fig. 1I and S6C) where optoMCP-FUS condensates, unlike optoFUS, clearly exhibit shape relaxation toward round morphologies. These data corroborate well with our statement of liquid-like behaviors of optoMCP-FUS condensates. The last sentence stating “this is actually confirmed later in figure 4” is unclear. In Fig. 4, we utilized the ternary phase behaviors of optogenetic systems to modulate the composition and material properties of co-condensates composed of both optoFUS and optoMCP-FUS. We note that in contrast to the optoMCP-FUS condensates, optoFUS showed solid gel-like behaviors upon light-activated assembly (Fig. S6B and C, PMID: 28041848).

* It seems that condensates behave differently with or without RNA (Fig. 1H D) . What would the explanation for this be? Could the mRNA somehow interfere with the condensation because in Fig. 1D there are single mRNAs without the MCP signal that appear the same size as 1 red puncta with RNA, while without RNA (Fig. 1D optoFUS) the mCherry dots look 2-3

fold larger. Could try activating the optoMCP-FUS longer to obtain "mature" condensates without MCP.

We found that the presence of cognate mRNAs promoted light-activated optoMCP-FUS phase separation, as evidenced by a decrease in the saturation concentration (Fig. 1F). This in turn led to higher total amount of the dense-phase, i.e., condensates, in the presence of cognate mRNAs (Fig. 1G and H). These behaviors are consistent with the known role of RNA in promoting phase separation (PMID: 26474065, 32632317), by acting as scaffolds to oligomerize phase-separating proteins. Regarding condensate size, we note that in general, the size of condensates dynamically changes, and is subject to multiple factors including activation time, the degree of super-saturation and etc. Moreover, the images of optoMCP-FUS condensates are often diffraction-limited, preventing the precise quantification of their sizes (further discussed in our answers below).

* Fig. 2B – line 162 – "we were able to capture the dynamics of the reporter mRNA recruitment into the condensates" – why is recruitment the term to describe what is seen. Rather, the mRNAs are there to begin with and the proteins that are bound to them aggregate and pull them along, or simply form around them. I think this experiment shows the formation of condensates that contain mRNAs within.

We thank the reviewer for raising this point. To monitor interactions between mRNAs and condensates as shown in Fig. 2, we used cells expressing high levels of optoMCP-FUS. In this regime of high optoMCP-FUS concentrations, optoMCP-FUS condensates can readily assemble through clustering of unbound species (Fig. 2B), rather than nucleating around the target mRNAs. This is because the degree of super-saturation can influence the nucleation patterns, and similar behaviors indeed have been observed in previous studies of light-activated condensates (PMID: 30500535, 35537448). However, we note that at this high level of expression, individual mRNAs are not distinguishable in the optoMCP-FUS channel (mCherry) due to high background. To further illustrate the effect of different expression levels on mRNA visualization and condensation, we included Fig. S2A in the revised manuscript. Here, we show fluorescence images of cells expressing three distinct levels of optoMCP-FUS. For direct visualization of individual optoMCP-FUS bound mRNAs, the background signals from unbound optoMCP-FUS molecules should be sufficiently low, a condition corresponding to the range of expressions much lower than the saturation concentration, C_{sat} , of optoMCP-FUS. In this regime, condensation does not take place even after light activation. For concentrations above C_{sat} , strong condensation is observed, yet individual mRNAs are not discernible. These behaviors motivated us to use the orthogonal labeling strategy for the target mRNA as in Fig. 2, where stdPCP-stdGFP is used for mRNA visualization. In this way, we were able to directly monitor individual mRNAs independent of their clustering.

*Fig. S2B – faster diffusion – but there are no diffusion measurements. One would need a diffusion coefficient to say something about diffusion properties. The figure shows that the mRNAs are freer to diffuse than the condensates. Looking at the movies in which the condensates don't seem to move very much and taking into account the FRAP interpretation above of a rigid structure, these structures are not really comparable. Also, the sentence "even though their sizes appear to be similar in the diffraction-limited images" has no real significance – one cannot infer such interpretation from them looking similar under a microscope. Finally, a control FRAP experiment of the condensate without mRNA should have

been performed to understand the effect of the RNA on the biophysical properties of the condensate.

In Fig. S2B, we showed mean-squared-displacement (MSD) analysis and diffusion coefficients for individual mRNAs and condensates, using data from cells expressing optoMCP-FUS at different levels (Fig. S2A). In the revised manuscript, we have also included MSD analysis results from the orthogonal labeling experiments (Fig. S3C). Diffusion coefficients of mRNAs are similar to the previously reported values (PMID: 30664789). What motivated us to perform MSD analysis is the apparent similarity between the size of mRNAs and that of condensates in fluorescence images. As the reviewer correctly mentioned, our point here is that condensates and mRNAs are highly distinct in terms of their mobilities.

Regarding the diffraction-limited imaging, our message may not be clearly delivered. When the size of object is smaller than the diffraction limit of optical microscopy (~ several hundreds of nm), the apparent size of the object in the image, limited by the point-spread-function, is bigger than the actual object size. Thus, the optical microscopy cannot reliably report size information of objects smaller than the diffraction limit. In our experiments of the dual labeling (Fig. 2 and S3), the sizes of condensates and individual mRNAs, monitored with mCh and GFP, respectively, were sometimes apparently similar. However, considering the limitation of diffraction-limited microscopy, this does not necessarily mean that their actual sizes are similar. Our MSD analysis is fully consistent with the expectation that condensates, composed of numerous proteins, are larger than individual mRNA molecules. We note that the absence of mCh signals within mRNA/GFP foci in Fig. 2B and E is due to high background of unbound optoMCP-FUS molecules which is needed to induce phase separation (Fig. S2A). Considering the reviewer's comment, we have updated the main text to better convey our message.

The FRAP data in the absence of cognate mRNAs are included below:

Fig. R1. Fluorescence recovery curves in the FRAP experiments of the MEF cells expressing optoMCP-FUS. Error bars, SD. n = 14 cells for MBS-KI MEF cells and n = 6 for WT MEF cells.

* Fig. 2E,F- Line 171 – "Strikingly" – what is so striking about them meeting and separating. Also, how frequent are these encounters – do all condensates acquire the mRNAs that way, because beforehand the authors wrote that the mRNAs act as oligomerization scaffolds, which would mean that they nucleate from the mRNA or together.

To monitor interactions between individual mRNAs and condensates, we chose cells with high expression of optoMCP-FUS. This experimental condition, leading to rapid induction of light-

mediated phase separation, was necessary due to the limited photostability of GFP and fast diffusive motions of mRNAs. As mentioned above, in this expression level, optoMCP-FUS condensation occurs through clustering of unbound species (Fig. 2B), leading to the formation of optoMCP-FUS condensates without containing the mRNAs, which later on begin to associate with the target mRNAs.

In Fig. 2, we employed two different image acquisition rates. A slow acquisition rate (1 frame/min) was used to capture the overall rate of mRNA accumulation, which happened over a few minutes timescale (Fig. 2B-D). We then used a faster rate (20 frames/sec) to monitor the recruitment processes of individual mRNAs. We note that, prior to the fast mode of image-acquisition, the transient nature of mRNA-condensate interactions was not necessarily expected during the mRNA recruitment process.

Lines 168-182: What do the authors suspect is the mechanism as to why some of the mRNAs tether tightly and others just graze the surface of the condensates. The authors suggest that the variance is due to translation capacity of the transcripts, but what would make some transcripts primed for translation and not others? Although many studies have documented that different mRNAs have different associations to various condensates, it appears there is significant variation among these identical constructs. One thing this reader was left wondering was what is the relationship between these condensates and overall translation of the cell. Does it affect one of the two canonical phosphorylation pathways that inhibit translation and yield stress granule condensates? (Should be mentioned that it has been shown in recent years that RNA in stress granules can be translationally active.) Although in figure 3 translational capacity was addressed, it seems relevant to understand general translation occurring at global cellular level, rather than the one transcript interacting directly with the condensate. These condensates form and disperse rapidly, but figure 3 demonstrates that the light can be administered for many hours. This questions how a huge number of large condensates affects translation, and not just one transcript.

We thank the reviewer for this suggestion. To find the effect of light-induced condensation on global translation at the cellular level, we have performed OP-Puro (O-propargyl-puromycin) labeling experiments (PMID: 22160674). The assay involves covalent labeling of nascent polypeptides with OP-Puro, which is then visualized via Click-chemistry-based fluorophore labeling. We found that blue-light activated condensation led to a minimal change in the global translation level, measured by the intensity of OP-Puro in individual cells (Fig. S5B). In the revised manuscript, we have also included data for stress granule formation. We confirm that stress granule is not induced by light-activated optoMCP-FUS/RNA condensation (Fig. S5A).

Regarding the observed heterogeneity in mRNA associations with optoMCP-FUS condensates, our result is consistent with a previous study (PMID: 302664789) showing a bimodal distribution of interaction times between mRNAs and condensates. What we meant by the translation status of mRNA is not the perpetual property of each mRNA transcript, but the status at the time of contact to condensates, which can vary over time for the given mRNA transcript. We have updated the manuscript to improve readability.

Also, in Fig. 2E it appears that the RNA can be bound or unbound or even associated (leaving and coming back), it would be interesting to measure a diameter for "associated" RNA and see if shifts between unbound-bound-associated can be measured over time. Namely can the 'saturation level' (line 167) change.

The contour length, i.e., the fully extended length, of the mRNA transcript used in Fig. 2 is ~ 800 nm (0.59 nm per base, PMID: 24411256). Our quantification of the distances between mRNAs and condensates during physical associations are within this limit (Fig. 2F and G). By saturation in line 167, we meant that after reaching the saturation level, the recruitment of mRNAs into light-activated optoMCP-FUS condensates does not change much over time.

* Fig. 3: More protein yields more aggregates which increases translation inhibition. This is then shown in the data in Fig. 3F under light activation. But a control is missing, of translation levels when no light is administered. What effects do high levels of this construct have on the transcripts, prior to formation of condensates. Otherwise, although it seems logical that the condensates directly impact on mRNA functionality and not just more RNA binding, this has to be shown. This point is also missed in the scheme in Fig. 3A, where it shows red condensates forming after addition of blue light, but no red whatsoever in the cells in the untreated state. This is of course not the case, and the FUS construct is free in the cell.

The schematic in Fig. 3A contains multiple features of different colors, so it is tough to include the presence of free optoMCP-FUS molecules in the cytoplasm. Regarding translation inhibition, we note that our quantification is based on the normalization in individual cells with respect to tagBFP intensities before blue light activation (Fig. 3B). However, considering the reviewer's comment, we have performed the control experiment to probe the effect of optoMCP-FUS binding to the 3'UTR of the reporter mRNA on translation (Fig. S4A and B). We found that cells expressing higher concentrations of optoMCP-FUS tended to exhibit higher tagBFP levels, implying that optoMCP-FUS binding may stabilize the reporter mRNA (PMID: 29131164). In the revised manuscript, we have included this data and updated the main text.

* The story shifts to an unclear path around figure 4. The analysis could be interesting as a physics examination but doesn't really add much to the story. Basically, the translation reduction was concluded without all this.

We again emphasize that our work goes beyond the simple demonstration of condensate-mediated translation inhibition. In Fig. 4, we aim to probe the relation between material property and function of condensates. It has been thought that the material properties of condensates would impact the dynamics and intermolecular interactions of residing components, which in turn would influence condensate functionalities. Indeed, this is a key idea underlying the liquid-to-solid transitions as a pathogenesis mechanism for neurodegenerative diseases. In Fig.4, utilizing the ternary phase behavior of optogenetic components, we were able to tune the composition and material properties of condensates containing target mRNAs. We found that mRNA translation activity is sensitive to the material properties of condensates.

* Fig. S4B- The purpose of the experiment of puro-PLA should be explained in more detail. Which puncta are being considered PLA-positive? The images are small and its unclear what is being counted. Does the b-actin antibody bind to an epitope at the very N-terminus of the protein? Also, what is the significance of the experiment. Probably more controls should be shown. The conclusion is that the translation inhibition occurs within 20 minutes, but would there be reason to think otherwise?

We apologize for not clearly conveying the message of puro-PLA experiments. In Fig. 3 and 4, we used the fluorescence level of tagBFP to monitor the translation status of the target mRNA. However, the response time of tagBFP depends on degradation rate (several hours), as shown in the anisomycin-treated case (now included as Fig. S7). Using the puro-PLA assay, we tested whether translation inhibition takes place at much shorter time scales. In the revised manuscript, we have updated main text to better convey our message, and also replaced the corresponding figure with images of better quality (Fig. S8B). The β -actin antibody used in puro-PLA assay binds to the N-terminus region of the protein (1-100 aa).

Minor:

* The authors show in Fig. 1 the optoFUS without MCP, however, a control of the construct without the FUS domain is necessary. The authors claim based on previous research that the non-RNA binding IDR of FUS causes the condensation after blue light, but what will the MCP-Cry2-mCherry do under blue light?

Without self-associating IDRs, mCherry-Cry2 exhibits poor clustering behaviors (PMID: 28041848). In our study, the goal of using the optogenetic construct is to control phase separation with light input.

* Fig. 1D: It will be preferable to show the entire cell together with the enlargement. This will give a better indication on what we are seeing. Also, are the condensates in the cytoplasm only, or also in the nucleus. In case there are also nuclear condensates, then in the b-actin MEF cells where active genes are observed in the nucleus, are these also associated with the condensates?

Also, can barely see mRNA FISH signal in the OptoFUS section. Suggest to indicate with arrows what is mRNA and what is not.

In the revised manuscript, we have updated Fig. 1D to include the zoom-out image of cells, and also indicated smFISH signals of mRNA with arrowheads. In the nucleus, optoMCP-FUS condensates are present, but identifying loci of active transcription is challenging due to optoMCP-FUS-mediated clustering of mRNAs.

* Fig. S1B: Should add the entire cell image with the enlargement.

We now show both the zoomed -in and -out images of optoMCP-FUS cells at three different expression levels in Fig. S2A.

* The authors addressed the lack of visible transcripts bound to the MCP-FUS, and this was thought to be owed to the high levels of expression. But actually in the movies provided (414494_0_video_7354989_rq623n) the expression level is quite low, and the condensates appear rapidly after the FUS begins to aggregate. In this case, where are the transcripts before blue light exposure. In theory they are bound to the coat protein before aggregation begins, and there is no issue of overexpression. The authors mention the tight binding of MCP to the MS2 loops.

Consistent with the phase separation mechanism, the light-activated clustering of optoMCP-FUS exhibits a concentration threshold, i.e. saturation concentration (Fig. 1F). We found that near and above the saturation concentration of optoMCP-FUS, individual mRNA puncta were not observable using optoMCP-FUS signals due to the high background from unbound species (Fig. S2A). This behavior is expected: MS2-based mRNA imaging requires enough binding of fluorescent-protein tagged MCP to the target mRNA, but, at the same time, not too high levels of background from the free MCP (PMID: 21356986). The suitable condition is typically achieved using the NLS-tagged version of MCP for the imaging of cytoplasmic mRNAs. In the revised manuscript, we have now illustrated the effect of different optoMCP-FUS levels on the mRNA visualization and condensation (Fig. S2A). At the very low level, individual reporter mRNAs are discernible as optoMCP-FUS puncta. However, at high expression levels where light-inducible clustering occurs, high background signals hamper visualization of individual mRNAs. This was exactly the reason why we employed the dual color imaging of optoMCP-FUS and PCP-GFP (Fig. 2), which enabled an orthogonal localization of mRNAs, independent of condensation.

We note that the movie mentioned by the reviewer shows a cell expressing optoMCP-FUS at the level well above saturation concentration where individual mRNAs cannot be identifiable due to the reason stated above. Moreover, the movie was taken with laser scanning confocal microscopy, and single-molecule mRNA imaging was conducted with the EMCCD-equipped wide-field microscope.

* Were other wavelengths of light tested, to see if condensates will form upon exposure to other wavelengths? Also, throughout the manuscript it seems that different wavelengths (wide filters) are used to excite the Cry2 system. This implies excitation around 488nm, the same as the GFP channel. Wouldn't the overlap of wavelengths cause a "blue" light excitation? For instance, in line 163 "Prior to blue light exposure, individual mRNAs are visible in the GFP channel..." wouldn't the GFP imaging cause the condensation of the MCP proteins and thus may affect results? In prior work of the authors they showed that different wavelengths can activate the Cry2 system. Suggest to add a little more detail about this, while mentioning why different times are used for activation in every experiment.

Cry2 can be activated by a broad range of wavelengths of light below 500 nm (PMID: 18988809). Thus, prior to experiments, we paid extra attention not to expose our sample to any types of blue light including bright-field illumination. We use the first image taken with GFP channel as a proxy for the condition corresponding to pre-activation. This is a reasonable approximation since blue-light dependent dynamics observed in the experiments, such as clustering or mRNA recruitment into condensates, are much slower than a single time frame (0.05-2 sec depending on experiments). Different activation methods were used depending on the type of experiments: the blue laser for mRNA-condensate interactions, and the bandpass-filtered bright-field for global activation such as smFISH experiments. The detailed activation conditions were provided in Methods and figure legends. We note that the deactivation timescale of Cry2 in the absence of blue light is about a few minutes (PMID: 24793453).

* Because there is such a dependence on mCherry-FUS expression in order to yield condensate formation, there is a question of how many protein/RNA complexes are found in this granule. From Fig. 1 we see that unbound mRNA in the cells expressing just optoFUS

have a certain intensity (green). By measuring this intensity, perhaps it is possible to calculate how many mRNPs are found at the condensate in the opto-MCP-FUS, and then also get an answer as to how many of these RNA/protein complexes are necessary for formation (because there is a direct relation between the protein and RNA) rather than the somewhat vague term 'high expression'. It might not be too difficult to calculate the number of mRNAs present, and thus imply the number of FUS in each condensate, especially given that these condensates seem uniform in size.

Thank you for this suggestion. In the revised manuscript, we have now included analysis results for the number mRNAs localized to individual condensates (Fig. S1A and B). Within condensates, the reporter mRNAs were heterogeneously distributed from 1 to 10 mRNAs/condensate.

* Line 158/161 – "proper levels" "suitable levels" – I suppose the authors mean the ideal levels of expression that yield clear signal over the background.

This comment is directly related to the ones we have answered above: suitable expression levels of optoMCP-FUS and stdPCP-stdGFP are required for light-activated clustering and single mRNA visualization, respectively (Fig. S2A).

* Line 169 - Explain 20 Hz. Please add some detail at least in the materials section.

We meant the acquisition rate for imaging (20 Hz = 20 frames/sec). We have updated our manuscript to add this information.

* Line 189 - why 12 hours of activation? Is 20 min not enough like anisomycin (which is not mentioned in the materials aside of PLA context). Wouldn't 12 hours of blue light be toxic to the cells? Should show cell survival analysis to see that the cells are healthy.

We performed 12 hours of activation due to the slow response rate of tagBFP as a translation reporter (Fig. S7). As the reviewer correctly pointed out, puromycin experiments showed that translation inhibition occurs as early as 20 min after light activation. We did not observe any apparent signs of toxicity in our 12-hour activation experiments. During the experiment, we observed multiple events of cell division as below, indicative of their healthy states.

Fig. R2. Confocal images of U2OS cells expressing tagBFP-DD-12x(MBS-PBS), NLS-stdPCP-stdGFP, optoMCP-FUS, and miRFP-optoFUS during 12 hours of blue-light activation. Arrowheads indicate cells undergoing divisions.

* Fig. 4G - looks like in the small squares that group 2 regains its signal faster and stronger than group 1, not like the graph below - why? Please add to every FRAP experiment the time of blue light activation, at least in the materials sections. Would a condensate immobile fraction change from using different times to activate? Please show FRAP at different times of light (5min/20min/12hours). Number of cells measured in FRAP experiments is low.

In Fig. 4G, the FRAP recovery graphs are consistent with images shown above (faster and stronger recovery in the group 2). In Methods, we have included blue light activation conditions used in FRAP experiments. We also included more data in FRAP experiments. We feel that performing FRAP experiments for different durations of blue light activation is beyond the scope of the paper.

Fig. 4 - would the RNA accumulate differently in every group? Please check with smFISH the different mRNA sequestration to each group, it would strengthen your point.

Thank you for the suggestion. We have performed smFISH experiments for cells in the Group 1 and 2, and added the result in the revised manuscript (Fig. S6G and H).

* Fig. S4B - It is known that expression levels of a certain RNA-binding protein can itself change granule abundance mRNA dynamics. This is especially relevant when in Fig. S4D it is shown that the levels of mCherry alone change PLA puncta. Therefore, the lowering of PLA puncta after condensate formation in Fig. S4B may be affected by the amount of FUS expressed before light activation. Therefore, an image of the cell pre-light exposure should be shown in order to confirm that the amount of FUS in each cell that are now being compared are identical.

We first note that puro-PLA experiments require fixation of cells, so it is impossible to acquire images for both conditions of before and after blue light activation from the same cells. Here, we started from a single population of optoMCP-FUS expressing cells, and applied to them blue light activation conditions (+/-) before assaying for Puro-PLA. Although individual cells may express different levels of optoMCP-FUS (as evidenced by the broad distribution of the Puro-PLA puncta number), our Puro-PLA assay clearly reported the population-level decrease in the production of nascent β -actin polypeptides upon blue-light dependent condensation (Fig. S8B and C).

* Movie S4- add another movie of the stabilized interaction

Materials - smFISH- add the probes sequences, and add to where the probe binds- the MS2 loops or the mRNA exons.

We have included the movie (Movie S5) and information on the probes (Supplementary table 1) in the revised manuscript.

* Plugin are used to reduce noise. Explain if these could affect the analysis. Line 525: what's the plugin's name? add "Bleach correction" plugin if it's that.

We have included the relevant information. We confirmed that the overall results were the same with or without the noise reduction.

* For making this study more insightful, one could tag a few different proteins such as translation initiation factors together with the Cry2 system and see if there is any change in translation.

We thank the reviewer for this suggestion, but we feel that implementing another type of synthetic condensates is outside the scope of the paper.

Reviewer #3 (Remarks to the Author):

the manuscript by Lee et al, describes an optogenetic approach to initiate phase transition of a mRNA by blue light induced RNA-protein complex formation. The approach described uses the well studied MS2 coat protein (and variants thereof) for binding to mRNA that were modified with cognate aptamer domains. To gain light sensitivity, a cry2 system for light-triggered homo dimerisation and multimerisation was applied.

The authors showed that upon multimerisation, hence phase transition of the mRNA in cells the translational activity of respective RNAs was diminished. This feature is somehow expected, as ribosomal access to mRNAs will be inhibited in condensates, at least sterically. The authors compare translational activity of condensed mRNA vs non-condensated mRNA. The authors extended their approach to endogenous proteins/mRNA (actin). However, I am wondering what is the impact of the MS2 coat protein RNA assembly on translational efficacy per se?

Other than that, I think the quality of the MS is very high and the findings are of broad interest. Therefore, recommend acceptance.

We thank the reviewer for the positive comments. To investigate the effect of optoMCP-FUS binding on mRNA translation, we measured the dependence of tagBFP levels on optoMCP-FUS expression in the absence of light-activated condensation (Fig. S4A and B). Cells expressing higher levels of optoMCP-FUS tend to exhibit higher tagBFP levels, suggesting that optoMCP-FUS binding to the 3'UTR of target mRNAs may affect their stability (PMID: 29131164). However, we note that for translational activity measurements, tagBFP levels were monitored in individual cells during light-activated condensation, and normalized to intensities prior to blue light activation (Fig. 3). Moreover, contrary to the absence of condensation, higher optoMCP-FUS concentrations led to stronger translation inhibition in light-activated conditions (Fig. 3F). Thus, the results from control experiments further strengthen our conclusion that mRNA sequestration into condensates drives translation inhibition. In the revised manuscript, we have updated the main text to accommodate these results.

REVIEWER COMMENTS

Reviewer #1 (Remarks to the Author):

I am satisfied with the authors' responses to my and the other reviewers' comments and recommend publication

Reviewer #3 (Remarks to the Author):

I have read the revised version of the manuscript and find that the authors have adequately addressed all the points raised by the reviewers. I therefore recommend acceptance in its present form.

Reviewer #4 (Remarks to the Author):

In this manuscript, Lee, Moon et al describe an optogenetic approach for promoting the formation of RNA-protein granules in cells. Since the manuscript has already been reviewed by three reviewers, I will restrict my questions to what I feel is the key issue that has not been resolved: Does mRNA association within granules inhibit translation?

Unfortunately, as the authors point out, the long-half life BFP makes it challenging to show a casual relationship between mRNAs being position in their optogenic granules and translation repression. During the 12 hour experiment, it would seem possible that other explanations (transcription inhibition, reduced mRNA export, decreased mRNA stability) could also explain the effect they see on BFP fluorescence levels. Importantly any changes in the material properties of the granules could similarly affect gene expression at a point other than translation. At this point, I think they can only conclude that overall gene expression is reduced but cannot definitely claim an inhibition of translation is the cause. Similar alternative explanations can also be made for the PLA experiment as well as the changes in spine volume. SunTag or other nascent peptide chain imaging experiments would be necessary in order to directly demonstrate translation inhibition of only mRNA that are within the opt-granules.

In my opinion though, it is not absolutely necessary for the authors to demonstrate translation inhibition for this manuscript to be published and I would strongly suggest modifying their interpretation of their data and allow for other possible explanations.

Response to Reviewers :

We thank the reviewers for their constructive comments and have revised the manuscript according to their suggestions. We provide a detailed point-by-point response to each of the reviewers' comments below (our responses are in blue text):

REVIEWER COMMENTS

Reviewer #1 (Remarks to the Author):

I am satisfied with the authors' responses to my and the other reviewers' comments and recommend publication

We thank the reviewer for recommending publication of our article.

Reviewer #3 (Remarks to the Author):

I have read the revised version of the manuscript and find that the authors have adequately addressed all the points raised by the reviewers. I therefore recommend acceptance in its present form.

We thank the reviewer for recommending acceptance of our study.

Reviewer #4 (Remarks to the Author):

In this manuscript, Lee, Moon et al describe an optogenetic approach for promoting the formation of RNA-protein granules in cells. Since the manuscript has already been reviewed by three reviewers, I will restrict my questions to what I feel is the key issue that has not been resolved: Does mRNA association within granules inhibit translation?

Unfortunately, as the authors point out, the long-half life BFP makes it challenging to show a casual relationship between mRNAs being position in their optogenic granules and translation repression. During the 12 hour experiment, it would seem possible that other explanations (transcription inhibition, reduced mRNA export, decreased mRNA stability) could also explain the effect they see on BFP fluorescence levels. Importantly any changes in the material properties of the granules could similarly affect gene expression at a point other than translation. At this point, I think they can only conclude that overall gene expression is reduced but cannot definitely claim an inhibition of translation is the cause. Similar alternative explanations can also be made for the PLA experiment as well as the changes in spine volume. SunTag or other nascent peptide chain imaging experiments would be necessary in order to directly demonstrate translation inhibition of only mRNA that are within the opt-granules.

In my opinion though, it is not absolutely necessary for the authors to demonstrate translation inhibition for this manuscript to be published and I would strongly suggest modifying their interpretation of their data and allow for other possible explanations.

We appreciate the reviewer for the suggestion. We understand that the reviewer has a concern that the reduced expression of BFP could be due to other possibilities such as reduced transcription, export, and stability of the target mRNA. To address this comment, we have added new figures (Fig. S8D and E) in the revised manuscript showing that no changes in the total abundance of target mRNAs were observed during 20-min of light activation. We also emphasized in the revised manuscript that the Puro-PLA assay clearly showed a decrease in the number of nascent polypeptides produced from the target mRNA as early as 20 min after light activation. Thus, it was the spatial redistribution of target mRNAs, rather than changes in the amount of them, that led to the observed decrease in protein expression in our system. However, we absolutely agree that nascent peptide chain imaging would provide further important insights into how mRNA association with condensates impacts translation output, especially kinetic aspects of the process. In the revised manuscript, we have added a discussion on the remaining questions that can be addressed using this powerful technique.

REVIEWERS' COMMENTS

Reviewer #4 (Remarks to the Author):

The authors have appropriately responded to my question and I support publication